# Dual Contrastive Inversion with Distributional Priors for Diversity-Aware Data-Free Knowledge Distillation

## Abstract

Model inversion (MI) has emerged as a key paradigm for data-free knowledge distillation (DFKD), yet existing MI methods suffer from limited diversity in synthetic data due to simplistic unimodal priors and the lack of explicit mechanisms for instance separability. We propose **D2CIP** (*Dual Contrastive Inversion with Distributional Priors*), a two-stage framework that enhances diversity by first recovering a class-conditional distributional prior with a Gaussian Mixture Model (GMM) aligned to teacher predictions and batch-normalization statistics, and then applying dual contrastive learning at both latent and instance levels with memory banks to enlarge the set of negatives. We further formalize data diversity as expected pairwise separability and establish its monotonic relationship with the contrastive loss, providing a principled justification for diversity maximization. Experiments on CIFAR-10, CIFAR-100, and Tiny-ImageNet demonstrate that D2CIP consistently outperforms state-of-the-art MI-based DFKD methods in both synthetic data diversity and distillation accuracy. The code is available at https://anonymous.4open.science/r/D2CIP4dfkd-53E3.

## 1 Introduction

Model inversion (MI) has gained significant attention as a technique for reconstructing data representations directly from trained models, supporting applications such as adversarial analysis Ho et al. (2024); Xia et al. (2025), model interpretability Cheng et al. (2024); Mahajan et al. (2024), and data-free knowledge distillation (DFKD) Shin & Choi (2024). In the context of DFKD, MI enables a student model to be trained without access to the original dataset by generating synthetic samples from a teacher model. However, existing MI-based DFKD methods Chen et al. (2019); Yin et al. (2020); Fang et al. (2021b); Binici et al. (2022); Shin & Choi (2024) often suffer from limited diversity in the generated data, which can lead to model collapse and poor generalization. This limitation primarily stems from two issues: (i) the reliance on simplistic unimodal priors (e.g., a single Gaussian) that fail to capture the inherently multi-mode structure of real data distributions, where different classes typically correspond to distinct modes; and (ii) the lack of explicit mechanisms to encourage separability among synthetic instances. Addressing these challenges is crucial for improving both the quality of synthetic data and the effectiveness of DFKD.

Recent studies have highlighted the central role of diversity in MI-based DFKD. Contrastive Model Inversion (CMI) Fang et al. (2021a) introduced contrastive learning to enforce instance-level discrimination, while Binici et al. Binici et al. (2022) leveraged a variational autoencoder to stabilize rehearsal without requiring data storage. TA-DFKD Shin & Choi (2024) further removes class-prior constraints to enhance sample diversity. Despite their progress, these approaches remain constrained by a diversity–stability trade-off: methods like CMI risk overfitting to dense regions, VAE-based approaches may smooth away fine-grained variations (PRE-DFKD), and filtering-based methods can discard challenging but valuable instances (TA-DFKD). Thus, there remains a gap in jointly achieving *expressive priors* and *principled diversity optimization*.

To address the limitations of existing MI-based DFKD methods, we propose **D2CIP** (*Dual Contrastive Inversion with Distributional Priors*), a two-stage framework explicitly designed to enhance the diversity of synthetic data. *Stage 1 (Distributional Prior Recovery)* learns a class-conditional

distributional prior, arameterized as a Gaussian Mixture Model (GMM), and optimized jointly with the generator by aligning with the teacher's predictions and internal batch-normalization statistics. Unlike unimodal or class-agnostic priors, this class-conditional distributional design captures the multi-mode structure of real data and provides a richer foundation for synthesis. *Stage 2 (Dual Contrastive Inversion)* then leverages this distributional prior by introducing contrastive objectives at both the latent and instance levels, supported by memory banks that enlarge the pool of negatives across inversion steps. Crucially, we establish a principled link between dataset diversity and contrastive learning by defining diversity as expected pairwise separability and demonstrating its monotonic relationship with the contrastive loss. This makes contrastive optimization a theoretically grounded surrogate for maximizing diversity. Finally, for downstream distillation, we adopt decision adversarial distillation to emphasize samples near the decision boundary, ensuring that the student benefits from both diverse and informative synthetic data.

The contributions of this paper can be summarized as follows: ❶ We propose a novel **D2CIP** framework that explicitly enhances the diversity of synthetic data for DFKD. By combining distributional priors with dual contrastive learning, D2CIP provides a principled and robust foundation for effective MI-based DFKD. ❷ We introduce a class-conditional distributional prior, modeled as a Gaussian Mixture Model (GMM), to address two key challenges: (1) aligning the learned latent distribution with the teacher model for consistency, and (2) capturing the inherently multi-mode structure of real data to better represent its complexity and diversity. ❸ We develop a dual contrastive inversion strategy that applies contrastive learning simultaneously at the latent and instance levels, with memory banks to enlarge the negative sample space. This design enriches the latent representation space and strengthens instance discrimination, thereby maximizing data diversity in a theoretically grounded manner. ❹ Extensive experiments on CIFAR-10, CIFAR-100, and Tiny-ImageNet validate the effectiveness of D2CIP, showing consistent improvements over state-of-the-art MI-based DFKD methods in both data diversity and distillation performance, with ablation studies confirming the contributions of distributional priors and dual contrastive objectives.

## 2 RELATED WORKS

Model inversion (MI) has recently been applied to data-free knowledge distillation (DFKD), where methods such as CMI Fang et al. (2021c) and SSD-KD Liu et al. (2024) enhance data diversity and distillation efficiency. Gaussian Mixture Models (GMMs) are widely used to model multimodal distributions and have shown effectiveness in tasks like domain adaptation Montesuma et al. (2024) and rare event modeling Li et al. (2024). In parallel, contrastive learning improves representation diversity and generalization, with recent works proposing diversity-aware objectives Author & Author (2023a) and uniformity-regularized strategies Zhou et al. (2023). These directions collectively highlight the importance of diversity in both data synthesis and representation learning. More detailed related works are provided in Appendix A.9.

## 3 METHOD

### 3.1 GENERATOR-BASED MODEL INVERSION FRAMEWORK

In the generator-based Model Inversion (gMI) framework, two key models are involved: a pre-trained target model $\theta_T$ and an auxiliary generator $\theta_G$. The main objective of gMI is to invert $\theta_T$ to recover a synthetic dataset $\mathcal{X}'$ that follows the same distribution as the original training dataset $\mathcal{X}$ used for $\theta_T$, with the assistance of $\theta_G$. Specifically, gMI seeks to jointly optimize the generator $\theta_G$ and its input noise vector $z \sim \mathcal{N}(\mu, \sigma^2)$ to generate a sample $x' = \theta_G(z)$, where $x' \in \mathcal{X}'$. This optimization aims to minimize the distance $d(\phi(x), \phi(x'))$, where $x \in \mathcal{X}$, $d$ is a distance metric (e.g., mean squared error), and $\phi(\cdot)$ represents a target mapping function, such as the student model $\theta_S$ in Data-Free Knowledge Distillation (DFKD) downstream tasks. Consequently, the primary challenge in current gMI methods lies in training $\theta_G$ under the guidance of $\theta_T$ without access to any prior information.

**General gMI Framework** To achieve robust performance in gMI, previous studies have enhanced the training of the generator from various perspectives. We summarize three key improvement considerations for generator training from the perspective of training loss, as follows:

- **Distributional Similarity** The first consideration is to enhance the distributional similarity between the original dataset $\mathcal{X}$ and the synthetic dataset $\mathcal{X}'$. Since direct access to $\mathcal{X}$ is unavailable in gMI, computing the true distributional disparity is infeasible. Instead, leveraging the fact that the generator typically performs class-conditional data generation by assigning a random category $y$ to each generated sample $x'$, we can measure the disparity between the predicted class label of $\theta_T$ for $x'$ and the assigned label $y$. This is achieved by introducing a class prior loss Chen et al. (2019):

$$\mathcal{L}_{cls}(x', y; \theta_G, \theta_T) = CE(\theta_T(x'), y), \tag{1}$$

where $CE(\cdot)$ denotes the cross-entropy loss.

- **Training Stability** To improve the training stability of the generator, a batch normalization (BN) loss is introduced by Yin et al. (2020); Fang et al. (2021a). This loss regularizes the feature distributions $\mathcal{N}(\mu_l(x'), \sigma_l^2(x'))$ of the layers in $\theta_G$ to align with the BN statistics $\mathcal{N}(\mu_l, \sigma_l^2)$ of the target model $\theta_T$. The BN loss is:

$$\mathcal{L}_{bn}(x'; \theta_G, \theta_T) = \sum_l \left( \| \mu_l(x') - \mu_l \|_2 + \| \sigma_l^2(x') - \sigma_l^2 \|_2 \right), \tag{2}$$

where $l$ denotes the number of feature layers in $\theta_G$, $\mu_l(x')$ and $\sigma_l^2(x')$ represent the batch-wise mean and variance of the feature outputs from the $l^{th}$ layer of $\theta_G$, and $\mu_l$ and $\sigma_l$ are the mean and variance of the $l^{th}$ feature layer of $\theta_T$. The term $|| \cdot ||_2$ denotes the $\ell_2$ norm operation.

- **Data Transferability** To better support downstream DFKD tasks, the generator in gMI should produce data samples that are close to the classification decision boundaries, thereby enhancing the transferability of the inverted data Micaelli & Storkey (2019). To achieve this, an adversarial distillation loss is employed to maximize the disagreement between the prediction logits of the target model $\theta_T(x')$ and the student model $\theta_S(x')$ Fang et al. (2019). This loss is defined as:

$$\mathcal{L}_{adv}(x'; \theta_G, \theta_T, \theta_S) = -KL(\theta_T(x'), \theta_S(x')), \tag{3}$$

where $KL(\cdot, \cdot)$ denotes the Kullback-Leibler (KL) divergence loss.

In summary, the three losses discussed above have been extensively demonstrated to enhance the robustness of gMI and effectively support the corresponding DFKD task Fang et al. (2021a). By combining these losses, a general gMI training loss function can be formulated as follows:

$$\mathcal{L}_{gMI} = \alpha \cdot \mathcal{L}_{cls} + \beta \cdot \mathcal{L}_{bn} + \gamma \cdot \mathcal{L}_{adv}, \tag{4}$$

where $\alpha$, $\beta$, and $\gamma$ are adjustable hyperparameters that control the weights of the respective losses.

**Limitations** While the current gMI training paradigm can achieve remarkable inversion for pre-trained models, the recovered synthetic data often struggles to meet the increasing demands for data quality in downstream DFKD tasks. Specifically, the recovered dataset $\mathcal{X}'$ must maintain the same distribution as the original dataset $\mathcal{X}$; otherwise, the student model in DFKD may **fail to distill** effective and accurate knowledge from out-of-distribution $\mathcal{X}'$. Additionally, sufficient class instance diversity in $\mathcal{X}'$ is crucial to prevent **student model collapse**, which arises from poor generalization on test data. However, these two critical DFKD concerns are not fully addressed in existing gMI studies, as evidenced by the following observations:

- **Simplistic Distributional Prior** Most existing gMI methods rely solely on a single distributional prior assumption, such as a Gaussian distribution, for the learnable generator $\theta_G$ to recover a single sample instance for a given class label during the inversion process. However, previous studies Yang et al. (2019); Liang et al. (2022) have shown that simple Gaussian distributions are insufficient to capture the complex modal representations of most real-world datasets, such as multi-class or multi-style vision data. Additionally, using a simple distributional prior for the noise vector $z$ can lead to mode collapse in the generated samples of $\theta_G$, which may subsequently cause the training collapse of student models in DFKD.

- **Low Instance Diversity** As discussed earlier, the use of a simple distributional prior for $\theta_G$ can implicitly reduce diversity of generated instances. Furthermore, the generator $\theta_G$ in gMI is expected to produce a variety of instances for a single class label, given the many-to-one mapping property of the student model $\theta_S$. However, existing training losses for $\theta_G$ do not incorporate any diversity constraints. On the contrary, the class prior loss, as shown in Eq. (1), may negatively impact both the quality and diversity of the generated samples, as highlighted in Shin & Choi (2024).

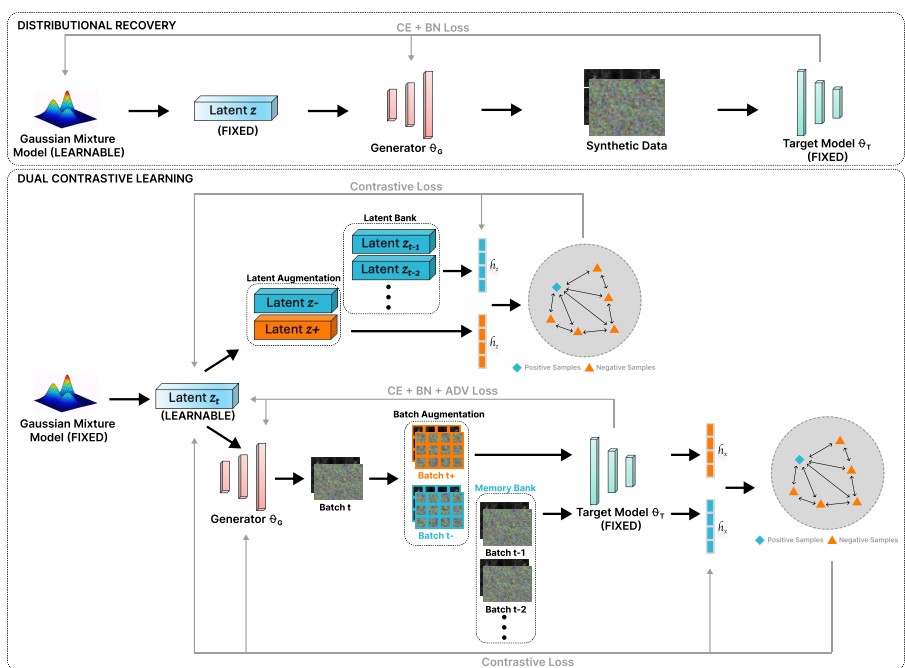

Figure 1: Illustration of the implementation process of *D2CIP*.

## 3.2 DUAL CONTRASTIVE INVERSION WITH DISTRIBUTIONAL PRIORS

**Overview** To overcome the limitations of existing gMI methods, we propose **D2CIP** (*Dual Contrastive Inversion with Distributional Priors*). As illustrated in Fig. 1, the framework comprises two stages: **Stage 1 (Distributional Prior Recovery)** leverages a class-conditional Gaussian Mixture Model (GMM) jointly optimized with the generator to recover expressive priors, while **Stage 2 (Dual Contrastive Inversion)** introduces contrastive objectives at both latent and instance levels, supported by memory banks, to enrich diversity in the synthesized data.

**Distributional Prior Recovery (Stage 1 of D2CIP)** For the generator training in gMI, the choice of distributional prior plays a crucial role in producing synthetic data that closely matches the distribution of the original data used to train the target model. Typically, the input vector $z$ for $\theta_G$ is sampled from a single Gaussian distribution $\mathcal{N}(\mu, \sigma^2)$. However, to address the limitations of a single Gaussian distribution in capturing complex data representations, we propose replacing the current single Gaussian prior assumption with a **class-conditional Gaussian Mixture Model (cGMM)** prior, inspired by Serban et al. (2017); Chen et al. (2021).

Specifically, for each class $y \in \{1, 2, \dots, C\}$, we employ a cGMM as the distributional prior for the input noise vector $z^{(y)}$ to $\theta_G$, defined as:

$$\mathcal{P}_{gmm}(z^{(y)} \mid y) = \sum_{k=1}^{K} \pi_k^{(y)} \cdot \mathcal{N}\big(z^{(y)} \mid \mu_k^{(y)}, \Sigma_k^{(y)}\big), \tag{5}$$

where $\pi_k^{(y)}$ is the weight of the $k^{th}$ Gaussian distribution for class $y$, and $\mu_k^{(y)}$ and $\Sigma_k^{(y)}$ are its mean and covariance, respectively, with $\sum_{k=1}^{K} \pi_k^{(y)} = 1$ and $\pi_k^{(y)} \geq 0$. Thus, $\{\pi_k^{(y)}, \mu_k^{(y)}, \Sigma_k^{(y)}\}_{k=1,2,\dots,K}$, along with $\theta_G$, are treated as learnable parameters estimated during the first stage of D2CIP. The training objective at this stage combines the class prior loss and the batch normalization (BN) loss:

$$\mathcal{L}_{s1}(x', y; \theta_G, \mathcal{P}_{gmm}) = \alpha \cdot \mathcal{L}_{cls} + \beta \cdot \mathcal{L}_{bn}. \tag{6}$$

To maintain an end-to-end differentiable optimization strategy for $\mathcal{P}_{gmm}$ and $\theta_G$, we adopt a reparameterization-based sampling scheme Kingma (2013). Specifically, we draw the latent vector by first sampling a component index $k \sim \text{Categorical}(\pi^{(y)})$, where the categorical distribution is parameterized by the class-dependent mixture weights $\pi^{(y)}$. In practice, this categorical distribution

can be instantiated in a uniform distribution when no prior knowledge is available. Once $k$ is selected, the latent vector is then sampled from the corresponding Gaussian component:

$$z^{(y)} = \mu_k^{(y)} + L_k^{(y)} \cdot \epsilon, \quad \epsilon \sim \mathcal{N}(0, I), \tag{7}$$

**Algorithm 1:** Distributional Prior Recovery (Stage 1 of D2CIP)

**Input:** Pre-trained target model $\theta_T$ and generator $\theta_G$
**Output:** cGMM prior $\mathcal{P}_{gmm}$

1 Initialize $\theta_G$ and $\mathcal{P}_{gmm}$;
2 **for** *number of update iterations* **do**
3     Sample class labels $y$;
4     Sample latent vector
      $z^{(y)} \sim \mathcal{P}_{gmm}(z \mid y)$ using Eq. (7);
5     Generate synthetic data $x' = \theta_G(z^{(y)})$;
6     Obtain target model predictions
      $y' = \theta_T(x')$;
7     Compute the loss $\mathcal{L}_{s1}$ using Eq. (6);
8     Update $\theta_G \leftarrow \theta_G - \eta \nabla_{\theta_G} \mathcal{L}_{s1}$;
9     Update
      $\mathcal{P}_{gmm} \leftarrow \mathcal{P}_{gmm} - \eta \nabla_{\mathcal{P}_{gmm}} \mathcal{L}_{s1}$;
10 **return** $\mathcal{P}_{gmm}$

where $L_k^{(y)}$ is the Cholesky decomposition of $\Sigma_k^{(y)}$ (i.e., $\Sigma_k^{(y)} = L_k^{(y)} L_k^{(y)T}$). For differentiability, the categorical sampling can be approximated by a Gumbel-Softmax relaxation during training, while at inference time we use standard discrete sampling. Once $\mathcal{P}_{gmm}$ is estimated, it is fixed and used as the distributional prior for generating $z^{(y)}$ in the second stage of D2CIP. The specific execution steps are detailed in **Algorithm 1**.

**Algorithm 2:** D2CIP (Dual Contrastive Inversion with Distributional Priors)

**Input:** Pre-trained target model $\theta_T$, generator $\theta_G$, class-conditional GMM prior $\mathcal{P}_{gmm}$, latent embedding header $h_z$, and synthetic data embedding header $h_x$
**Output:** Synthetic data bank $\mathcal{B}_x$

1 $\mathcal{B}_x \leftarrow \emptyset$;
2 $\mathcal{B}_z \leftarrow \emptyset$;             // Latent bank
3 Estimate $\mathcal{P}_{gmm}$;     // See Algorithm 1
4 **for** *each batch* **do**
5     Initialize $\theta_G$;
6     Sample class labels $y$;
7     Sample $z^{(y)} \sim \mathcal{P}_{gmm}(z \mid y)$ using Eq. (7);
8     **for** *number of update iterations* **do**
9        Compute $\mathcal{L}_{zcr}(z^{(y)} \cup \mathcal{B}_z, h_z)$ using
          Eq. (10);
10        $x' \leftarrow \theta_G(z^{(y)})$;
11        Compute $\mathcal{L}_{scr}(x' \cup \mathcal{B}_x, h_x)$ using
          Eq. (10);
12        Compute $\mathcal{L}_{s2\text{-}z}$ using Eq. (13);
13        Compute $\mathcal{L}_{s2\text{-}G}$ using Eq. (14);
14        Update $\theta_G \leftarrow \theta_G - \eta \nabla_{\theta_G} \mathcal{L}_{s2\text{-}G}$;
15        Update $h_x \leftarrow h_x - \eta \nabla_{h_x} \mathcal{L}_{s2\text{-}G}$;
16        Update $z^{(y)} \leftarrow z^{(y)} - \eta \nabla_{z^{(y)}} \mathcal{L}_{s2\text{-}z}$;
17        Update $h_z \leftarrow h_z - \eta \nabla_{h_z} \mathcal{L}_{s2\text{-}z}$;
18     $\mathcal{B}_z \leftarrow \mathcal{B}_z \cup z^{(y)}$;
19     $\mathcal{B}_x \leftarrow \mathcal{B}_x \cup x'$;
20 **return** $\mathcal{B}_x$

**Data Diversity via Contrastive Learning** Before proceeding to the second stage of D2CIP, we first discuss how to define data diversity. For a given pair of data instances $\{x'_1, x'_2\} \in \mathcal{X}'$, we can employ a distance comparison metric $d(\cdot, \cdot)$ to quantify their distinguishability. Consequently, the data diversity of $\mathcal{X}'$ can be measured by:

$$\mathcal{L}_{div}(\mathcal{X}') = \mathbb{E}_{x'_1, x'_2 \in \mathcal{X}'} \left[ d(x'_1, x'_2) \right]. \tag{8}$$

In contrastive learning Chen et al. (2020), data instance-level distinguishability is learned by treating each instance as a separate category in a self-supervised manner, providing an ideal technical approach to define $d(\cdot, \cdot)$. Inspired by Fang et al. (2021a), the pre-trained target model $\theta_T$ is employed as the basic feature representation extraction network, denoted as $\theta_T^f(\cdot)$. This is followed by a learnable embedding header $h(\cdot)$, which projects data instances into a new representational space via the mapping function $H(\cdot) = (h \circ \theta_T^f)(\cdot)$. Following the standard contrastive learning process, the cosine similarity is used to measure the similarity between $x'_1$ and $x'_2$ in this newly established space, formalized as follows:

$$sim(x'_1, x'_2, H) = \frac{\langle H(x'_1), H(x'_2) \rangle}{\|H(x'_1)\| \cdot \|H(x'_2)\|}. \tag{9}$$

In the contrastive learning framework, each instance $x'$ is randomly transformed into multiple diverse views. Typically, a positive view instance $x'^+$, which should closely match $x'$, is created, while all other view instances are treated as negative ones $x'^-$. Thus, the final optimization loss in contrastive learning is formalized as:

$$\mathcal{L}_{cr}(\mathcal{X}', H) = -\mathbb{E}_{x'_i \in \mathcal{X}'} \left[ \log \frac{\exp(sim(x'_i, x'^+_i, H)/\tau)}{\sum_j \exp(sim(x'_i, x'^-_j, H)/\tau)} \right], \tag{10}$$

where $\tau$ is the temperature parameter. From Eq. (10), we observe that $\mathcal{L}_{cr}$ aims to push positive pairs closer together while pulling negative pairs apart, achieving the objective of instance-level discrimination. If we define $d(x'_1, x'_2)$ as:

$$
\begin{aligned}
\mathcal{L}_{div}(\mathcal{X}') &= -\mathbb{E}_{x'_i, x'_j \in \mathcal{X}'} \left[ \frac{\exp(\mathrm{sim}(x'_i, {x'_j}^-, H)/\tau)}{\exp(\mathrm{sim}(x'_i, {x'_i}^+, H)/\tau)} \right] \\
&= -\frac{1}{C(x'^-)} \mathbb{E}_{x'_i \in \mathcal{X}'} \left[ \frac{\sum_j \exp(\mathrm{sim}(x'_i, {x'_j}^-, H)/\tau)}{\exp(\mathrm{sim}(x'_i, {x'_i}^+, H)/\tau)} \right] = \frac{1}{C(x'^-) \cdot \mathcal{L}_{cr}(\mathcal{X}', H)}.
\end{aligned}
\tag{11}
$$

where $C(x'^-)$ denotes the total number of negative instances for $x'_i$. Eq. (11) clearly demonstrates the inverse relationship between data diversity and contrastive loss, indicating that minimizing the contrastive loss $\mathcal{L}_{cr}$ maximizes the data diversity $\mathcal{L}_{div}$.

**Dual Contrastive Inversion (Stage 2 of D2CIP)** Building on the established connection between data diversity and contrastive learning, we now integrate contrastive learning into the second stage of D2CIP to further improve the diversity of the synthetic dataset $\mathcal{X}'$ generated by the generator. This is achieved through a dual contrastive learning-driven gMI training mechanism, as illustrated in Fig. 1.

Specifically, contrastive learning is applied to both the input latent vector $z^{(y)}$ and the output synthetic data $x' = \theta_G(z^{(y)})$. Since gMI follows an instance-wise inversion process, the generator synthesizes only one batch of data at each inversion timestamp $t$. At the beginning of timestamp $t$, $\theta_G$ is re-initialized, and a class label $y$ is first sampled; a learnable latent vector $z^{(y)} \sim \mathcal{P}_{gmm}(z \mid y)$ is then drawn from the class-conditional prior as the initial input to $\theta_G$. Subsequently, $z^{(y)}$ and $\theta_G$ are iteratively optimized.

In addition to the general inversion losses described in Eq. (4), two contrastive objectives are introduced: the latent contrastive loss $\mathcal{L}_{zcr}$ and the synthetic contrastive loss $\mathcal{L}_{scr}$. For computing $\mathcal{L}_{zcr}$, each latent vector $z_t^{(y)}$ is augmented by adding Gaussian perturbations to form a positive latent $(z_t^{(y)})^+$ and a negative latent $(z_t^{(y)})^-$. To further enrich the pool of negatives, a latent memory bank $\mathcal{B}_z = \{z_{t-1}^{(\cdot)}, z_{t-2}^{(\cdot)}, \dots\}$ is maintained, where each historical latent serves as a negative instance. Similarly, for computing $\mathcal{L}_{scr}$, each synthetic instance $x'_t$ is randomly transformed into ${x'_t}^+$ and ${x'_t}^-$, and a memory bank $\mathcal{B}_x = \{x'_{t-1}, x'_{t-2}, \dots\}$ stores previous synthetic samples to provide additional negatives. By contrasting current samples against historical ones, the memory banks guide the generator to produce more diverse batches.

The second-stage optimization objective is formulated as:

$$
\min_{\theta_G, z^{(y)}, h_z, h_x} \left[ \lambda_1 \mathcal{L}_{scr}(\theta_G(z^{(y)}) \cup \mathcal{B}_x, h_x) + \lambda_2 \mathcal{L}_{gMI}(\theta_G(z^{(y)})) + \lambda_3 \mathcal{L}_{zcr}(z^{(y)} \cup \mathcal{B}_z, h_z) \right], \tag{12}
$$

where $h_x$ and $h_z$ denote the projection heads for synthetic and latent contrastive learning, respectively. Note that $z^{(y)}$ and $\theta_G$ are updated with different objectives, and the update loss for $z^{(y)}$ is defined as:

$$
\mathcal{L}_{s2\text{-}z} = \lambda_2 \cdot \mathcal{L}_{gMI}(\theta_G(z^{(y)})) + \lambda_3 \cdot \mathcal{L}_{zcr}(z^{(y)} \cup \mathcal{B}_z, h_z), \tag{13}
$$

while the update loss for $\theta_G$ is:

$$
\mathcal{L}_{s2\text{-}G} = \lambda_1 \cdot \mathcal{L}_{scr}(\theta_G(z^{(y)}) \cup \mathcal{B}_x, h_x) + \lambda_2 \cdot \mathcal{L}_{gMI}(\theta_G(z^{(y)})). \tag{14}
$$

The complete procedure of D2CIP is summarized in Algorithm 2. The synthetic data accumulated in the memory bank $\mathcal{B}_x$ will be used for downstream DFKD tasks. A more detailed theoretical justification of these objectives, including analysis of diversity, separability, mode coverage, convergence, and generalization, is provided in Appendix A.2.

### 3.3 D2CIP-DRIVEN DFKD

With the synthetic dataset $\mathcal{X}'$ generated in D2CIP, we perform a DFKD task to train a student model $\theta_S$ from the teacher $\theta_T$ by employing a KL distillation loss. However, since $\mathcal{X}'$ cannot fully match the richness of real data, relying solely on KL divergence may lead to suboptimal student representations. To alleviate this limitation, adversarial distillation is commonly employed in DFKD, where the generator is guided to produce samples that maximize teacher–student disagreement.

Following Fang et al. (2021a), we adopt the decision adversarial variant, which restricts adversarial distillation to samples on which the teacher and student make consistent predictions. Formally, the decision adversarial distillation loss is defined as:

$$\mathcal{L}_{adv}(\mathcal{X}'; \theta_T, \theta_S) = -\mathbb{1}(\arg\max \theta_T(x') = \arg\max \theta_S(x')) \cdot KL\left(\frac{\theta_T(x')}{\tau} \,\Big\|\, \frac{\theta_S(x')}{\tau}\right), \quad (15)$$

where $\mathbb{1}(\cdot)$ is an indicator function. Eq. (15) encourages effective distillation near the decision boundary, while guiding the generator to synthesize informative and discriminative samples.

## 4 EXPERIMENT

### 4.1 DATASETS AND MODELS

We follow recent SOTA DFKD works Fang et al. (2021b); Binici et al. (2022); Yu et al. (2023); Shin & Choi (2024) and evaluate D2CIP on widely adopted backbones including ResNet He et al. (2016), VGG Simonyan (2014), and Wide ResNet Zagoruyko (2016). For benchmarking, we use three standard datasets: CIFAR-10 Krizhevsky et al. (2009), CIFAR-100 Krizhevsky et al. (2009), and Tiny-ImageNet Le & Yang (2015), ensuring comparability with prior studies. Detailed dataset and model configurations are provided in Appendix A.11.

### 4.2 BASELINES AND EVALUATION PROTOCOL

We compare D2CIP with representative DFKD methods, including DAFL Chen et al. (2019), ZSKT Nayak et al. (2019), DFQ Nagel et al. (2019), ADI Yin et al. (2020), CMI Fang et al. (2021b), PRE-DFKD Binici et al. (2022), and TA-DFKD Shin & Choi (2024). To evaluate effectiveness, we report the top-1 accuracy of the student model distilled on the generated dataset. In addition, Fréchet Inception Distance (FID) and Jensen-Shannon (JS) divergence are employed to assess the quality and diversity of the generated data. Further evaluation metrics and detailed quantitative results are provided in Appendix A.5, and hyperparameter settings are summarized in Appendix A.8.

### 4.3 PERFORMANCE OVERVIEW

We evaluate D2CIP on CIFAR-10, CIFAR-100, and Tiny-ImageNet under diverse teacher–student ($\mathcal{T}$–$\mathcal{S}$) settings, with results summarized in Table 1. On CIFAR-10, D2CIP achieves 95.16% in the $\mathcal{T}(a)$–$\mathcal{S}(b)$ setting and 93.70% in the $\mathcal{T}(d)$–$\mathcal{S}(f)$ case, both approaching teacher accuracy. This gain reflects the ability of our dual contrastive mechanism to preserve informative latent features while enhancing data diversity. On CIFAR-100, D2CIP obtains 71.66% for $\mathcal{T}(c)$–$\mathcal{S}(b)$, surpassing all baselines and showcasing its robustness in handling complex label spaces. On Tiny-ImageNet, despite the large number of classes and higher variability, it still achieves the highest accuracy of 61.49%, highlighting the scalability brought by distributional recovery in capturing richer data priors. To further assess the scalability of D2CIP on large-scale benchmarks, we additionally report experiments on the ImageNet-100 dataset in Appendix A.3. Overall, D2CIP leverages distributional prior recovery and dual contrastive learning to narrow the student–teacher gap while ensuring strong generalization.

### 4.4 EVALUATION OF DIVERSITY AND DISTRIBUTIONAL ALIGNMENT

We assess the effectiveness of **D2CIP** in enhancing synthetic data diversity through quantitative metrics. As shown in Fig. 2, D2CIP achieves the best performance on CIFAR-10 in terms of JS divergence and FID, reflecting superior distributional alignment and visual fidelity. Specifically, it attains the lowest JS divergence (0.498) and FID (110.56), outperforming CMI (0.531, 112.32) and TA-DFKD (0.569, 219.53). In addition, D2CIP exhibits lower class-wise variability, maintaining stable alignment across categories such as "deer" and "bird." By contrast, TA-DFKD shows unstable behavior with large divergence fluctuations, while CMI yields moderate yet inferior results.

Visualization results in Fig. 3 further highlight the advantages of D2CIP. For the representative "bird" and "horse" classes, D2CIP produces the highest overlap between real and synthetic data distributions, supported by the lowest JS divergence (0.50 and 0.47) and FID (73.16 and 79.62). This indicates strong semantic consistency and improved intra-class diversity. In contrast, TA-DFKD yields clearly separated clusters with large divergence, while CMI shows partial overlap but weaker

Table 1: Performance comparison of DFKD methods across multiple Teacher–Student ($\mathcal{T} - \mathcal{S}$) pairs. Teacher/Student models include (a) ResNet-34, (b) ResNet-18, (c) VGG-11, (d) WRN-40-2, (e) WRN-16-1, (f) WRN-40-1, and (g) WRN-16-2. A dash "–" indicates results either not reported in the original paper or not reproducible under the experimental setting, and **bold** highlights the best performance within each configuration.

| Method | CIFAR-10 | | | | | CIFAR-100 | | | | | Tiny-ImageNet |
|---|---|---|---|---|---|---|---|---|---|---|---|
| | $\mathcal{T}$(a) $\mathcal{S}$(b) | $\mathcal{T}$(c) $\mathcal{S}$(b) | $\mathcal{T}$(d) $\mathcal{S}$(e) | $\mathcal{T}$(d) $\mathcal{S}$(f) | $\mathcal{T}$(d) $\mathcal{S}$(g) | $\mathcal{T}$(a) $\mathcal{S}$(b) | $\mathcal{T}$(c) $\mathcal{S}$(b) | $\mathcal{T}$(d) $\mathcal{S}$(e) | $\mathcal{T}$(d) $\mathcal{S}$(f) | $\mathcal{T}$(d) $\mathcal{S}$(g) | $\mathcal{T}$(a) $\mathcal{S}$(b) |
| Teacher Model ($\mathcal{T}$) | 95.70 | 95.20 | 94.87 | 94.87 | 94.87 | 78.05 | 77.10 | 75.83 | 75.83 | 75.83 | 63.27 |
| Student Model ($\mathcal{S}$) | 95.20 | 92.25 | 91.12 | 93.94 | 93.95 | 77.10 | 71.32 | 65.31 | 72.19 | 73.56 | 61.37 |
| DAFL (ICCV'19) | 92.22 | 81.10 | 65.71 | 81.33 | 81.55 | 74.47 | 57.29 | 22.50 | 34.66 | 40.00 | – |
| ZSKT (ICML'19) | 93.32 | 89.46 | 83.74 | 86.07 | 89.66 | 67.74 | 34.72 | 30.15 | 29.73 | 28.44 | – |
| DFQ (ICCV'19) | 94.61 | 90.84 | 86.14 | 91.69 | 92.01 | 77.01 | 68.32 | 54.77 | 61.92 | 59.01 | 59.49 |
| ADI (CVPR'20) | 93.26 | 90.36 | 83.04 | 86.85 | 89.72 | 61.32 | 54.13 | 53.77 | 61.33 | 61.34 | – |
| CMI (IJCAI'21) | 94.84 | 91.13 | 90.01 | 92.52 | 92.78 | 77.10 | 70.56 | 57.91 | 68.88 | 68.75 | 60.56 |
| PRE-DFKD (AAAI'22) | 94.10 | 86.86 | 90.34 | 91.54 | 88.46 | 77.10 | 66.81 | 54.62 | 55.52 | 46.79 | 54.20 |
| TA-DFKD (AAAI'24) | 94.43 | 89.44 | 85.92 | 86.27 | 86.12 | 76.99 | 62.92 | 41.32 | 40.92 | 41.63 | 20.03 |
| **D2CIP (Ours)** | **95.16** | **91.70** | **90.75** | **93.70** | **93.48** | **77.36** | **71.66** | **59.69** | **70.03** | **69.85** | **61.49** |

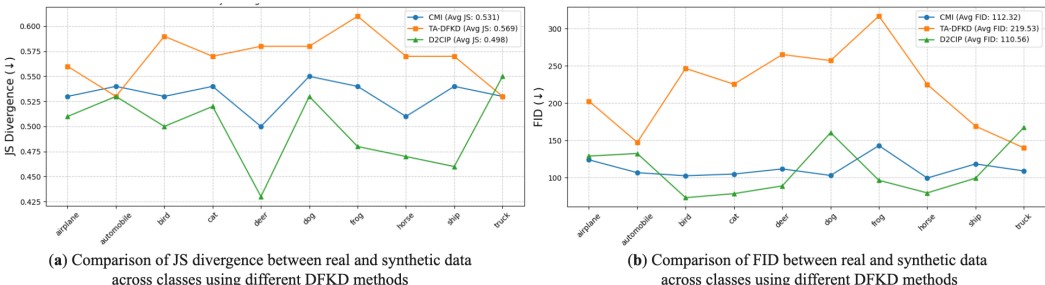

(a) Comparison of JS divergence between real and synthetic data across classes using different DFKD methods

(b) Comparison of FID between real and synthetic data across classes using different DFKD methods

Figure 2: Quantitative comparison of data generation diversity among CMI, TA-DFKD, and D2CIP using JS divergence and FID across classes on CIFAR-10. (a) JS divergence comparison, where the average values for CMI, TA-DFKD, and D2CIP are 0.531, 0.569, and **0.498**, respectively. (b) FID comparison, with corresponding average values of 112.32, 219.53, and **110.56**. The symbol ↓ indicates that lower values are preferred.

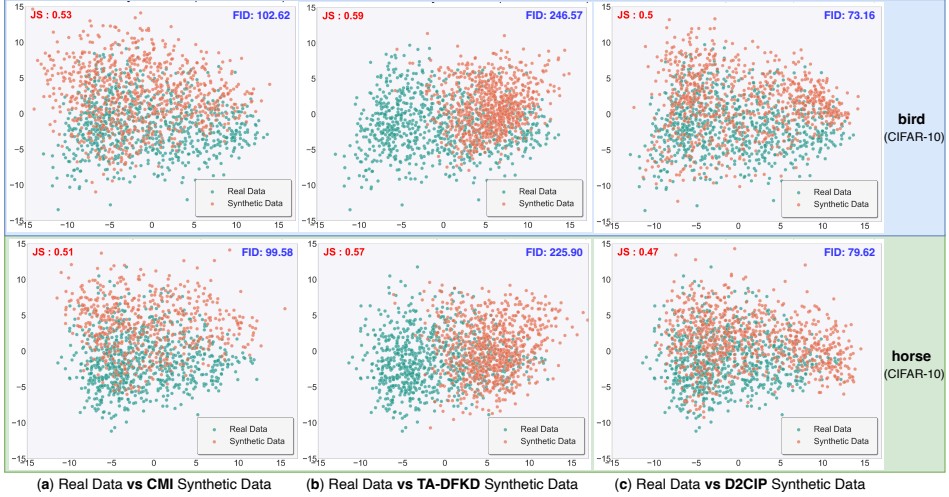

(a) Real Data **vs CMI** Synthetic Data    (b) Real Data **vs TA-DFKD** Synthetic Data    (c) Real Data **vs D2CIP** Synthetic Data

Figure 3: Distribution visualization of real and synthetic data using a two-dimensional mapping from a well-trained ResNet34 on CIFAR-10. (a) Real vs CMI-generated data for bird (top) and horse (bottom). (b) Real vs TA-DFKD-generated data for bird (top) and horse (bottom). (c) Real vs D2CIP-generated data for bird (top) and horse (bottom).

distributional fidelity. Taken together, these quantitative and qualitative results confirm that combining class-conditional GMM priors with dual contrastive optimization enables D2CIP to generate synthetic data with both high diversity and strong visual realism. Additional qualitative results are provided in Appendix A.5.3 and Appendix A.7.

## 4.5 ROBUSTNESS TO cGMM HYPERPARAMETERS

We analyze the effect of the number of cGMM components ($K$) and mixture weight strategies ($\pi_k$) on the performance of **D2CIP**. As shown in Table 2, the overall performance remains relatively stable as $K$ varies, with only minor fluctuations across different settings. On CIFAR-10, learnable $\pi_k$ achieves slightly higher results in some cases (e.g., **95.16%** at $K = 5$ for $\mathcal{T}(a)$-$\mathcal{S}(b)$), while on CIFAR-100, fixed uniform weights sometimes perform better (e.g., **77.36%** at $K = 5$ for $\mathcal{T}(a)$-$\mathcal{S}(b)$ and **70.03%** for $\mathcal{T}(d)$-$\mathcal{S}(f)$). For Tiny-ImageNet, both strategies deliver comparable results, with the best accuracy of **61.49%** at $K = 2$ under learnable weights. These observations suggest that although different $K$ and $\pi_k$ settings may yield slight gains, D2CIP maintains consistently high performance without large swings, underscoring its insensitivity to hyperparameter variations.

Table 2: Performance variance with varying $K$ values under two different weight combination strategies (learnable $\pi_k$ and fixed $\pi_k$) for GMM components in D2CIP. Teacher Models ($\mathcal{T}$) and Student Models ($\mathcal{S}$) are selected from (a) resnet-34, (b) resnet-18, (c) vgg-11, (d) wrn-40-2, (e) wrn-16-1, (f) wrn-40-1, (g) wrn-16-2.

| Dataset | $\mathcal{T}$ | $\mathcal{S}$ | With learnable $\pi_k$ | | | | | With fixed $\pi_k = 1/K$ | | | | |
|---|---|---|---|---|---|---|---|---|---|---|---|---|
| | | | $K=1$ | $K=2$ | $K=3$ | $K=4$ | $K=5$ | $K=1$ | $K=2$ | $K=3$ | $K=4$ | $K=5$ |
| CIFAR-10 | $\mathcal{T}(a)$ | $\mathcal{S}(b)$ | 94.89 | 94.68 | 94.97 | 94.82 | **95.16** | 94.79 | 94.66 | 94.89 | 94.74 | **95.00** |
| | $\mathcal{T}(c)$ | $\mathcal{S}(b)$ | 91.22 | 90.95 | 91.12 | **91.24** | 90.91 | **91.70** | 91.42 | 91.38 | 91.68 | 91.59 |
| | $\mathcal{T}(d)$ | $\mathcal{S}(e)$ | 89.82 | 89.56 | 89.89 | **89.97** | 89.75 | 89.75 | 89.82 | **90.75** | 89.24 | 88.88 |
| | $\mathcal{T}(d)$ | $\mathcal{S}(f)$ | 93.37 | **93.70** | 93.64 | 92.37 | 93.43 | 93.33 | **93.49** | 92.44 | 92.43 | 93.31 |
| | $\mathcal{T}(d)$ | $\mathcal{S}(g)$ | 92.75 | 93.23 | **93.32** | 93.04 | **93.32** | 92.80 | 93.22 | **93.48** | 92.94 | 93.13 |
| CIFAR-100 | $\mathcal{T}(a)$ | $\mathcal{S}(b)$ | 76.48 | 76.33 | 76.17 | 76.16 | **76.59** | 76.96 | 76.04 | 76.78 | 76.96 | **77.36** |
| | $\mathcal{T}(c)$ | $\mathcal{S}(b)$ | 71.31 | 71.55 | **71.66** | 70.19 | 70.31 | 71.30 | **71.38** | 70.13 | 70.25 | 71.25 |
| | $\mathcal{T}(d)$ | $\mathcal{S}(e)$ | 58.81 | 58.69 | **58.88** | 57.93 | 58.78 | 59.57 | 59.29 | 58.67 | 58.35 | **59.69** |
| | $\mathcal{T}(d)$ | $\mathcal{S}(f)$ | **69.58** | 68.57 | 67.95 | 68.38 | 68.61 | 69.98 | 68.86 | 68.36 | 68.97 | **70.03** |
| | $\mathcal{T}(d)$ | $\mathcal{S}(g)$ | 69.90 | 69.54 | **70.58** | 68.61 | 68.32 | 69.33 | 68.74 | 68.90 | 69.01 | **69.85** |
| Tiny-ImageNet | $\mathcal{T}(a)$ | $\mathcal{S}(b)$ | 61.23 | **61.49** | 60.96 | 61.17 | 61.24 | 61.12 | 60.80 | 60.25 | **61.41** | 61.12 |

## 4.6 ABLATION STUDY

We perform an ablation study to examine the contribution of each component in **D2CIP**. As shown in Table 3, removing the instance-level contrastive loss $\mathcal{L}_{scr}$ or the latent-level loss $\mathcal{L}_{zcr}$ consistently reduces accuracy, confirming their complementary roles in promoting discriminative synthetic data and well-structured latent spaces. The largest drop occurs when discarding the cGMM prior, which lowers accuracy to **94.17%** on CIFAR-10 and **73.58%** on CIFAR-100, highlighting its central importance for distributional diversity. These results demonstrate that both contrastive objectives and the GMM-based prior are indispensable to the strong distillation performance of D2CIP. Additional sensitivity studies on loss weights and the contrastive temperature are reported in Appendix A.4.

Table 3: Top-1 accuracy on CIFAR-10/100 with a ResNet-34 teacher and a ResNet-18 student.

| Method | CIFAR-10 | CIFAR-100 |
|---|---|---|
| D2CIP | **95.16** | **77.36** |
| w/o $\mathcal{L}_{scr}$ | 94.44 | 76.23 |
| w/o $\mathcal{L}_{zcr}$ | 94.83 | 75.90 |
| w/o cGMM | 94.17 | 73.58 |

Beyond these evaluations, we also provide a detailed analysis of the theoretical and empirical computational complexity of D2CIP in Appendix A.6.

## 5 CONCLUSION

We proposed *Dual Contrastive Inversion with Distributional Priors (D2CIP)* to enhance data diversity in Data-Free Knowledge Distillation (DFKD). By integrating class-conditional GMM-based distributional priors with dual contrastive learning at latent and instance levels, D2CIP produces synthetic data that are both diverse and semantically aligned, enabling more effective knowledge transfer. Experiments on CIFAR-10, CIFAR-100, and Tiny-ImageNet show consistent improvements over state-of-the-art baselines in both diversity and distillation performance. Limitations of D2CIP are further discussed in Appendix Section A.10.

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

# A APPENDIX

## A.1 THE USE OF LARGE LANGUAGE MODELS

The language of this paper was polished using large language models (LLMs) to enhance clarity and readability. The final content and academic integrity remain the responsibility of the authors.

## A.2 EXTENDED THEORETICAL ANALYSIS FOR D2CIP

This section provides an extended theoretical justification for why the proposed *Dual Contrastive Inversion with Distributional Priors (D2CIP)* improves synthetic data diversity and distillation performance.

### A.2.1 INFONCE LOSS AND DATA DIVERSITY

The instance-level contrastive objective in Eq. (10) is based on the InfoNCE loss. For each anchor $x_i$, there is a positive sample $x_i^+$ and $M$ negative samples $\{x_j^-\}_{j=1}^M$. The temperature $\tau > 0$ controls the sharpness of similarity weighting, and $\text{sim}(\cdot, \cdot)$ denotes cosine similarity.

The core idea is that the model should always place an anchor closer to its positive than to any negative. The separability margin is defined as

$$\Delta = \text{sim}(x_i, x_i^+) - \tfrac{1}{M} \sum_{j=1}^{M} \text{sim}(x_i, x_j^-), \tag{16}$$

which quantifies how much better positives are aligned compared to negatives. A simple inequality shows that minimizing the InfoNCE loss or enlarging $M$ increases $\Delta$. Thus the contrastive loss is not only a training signal, but also a formal surrogate for maximizing diversity, as already suggested in Eq. (11).

A larger $M$ means each anchor competes against more negatives, reducing mode collapse. A smaller $\tau$ emphasizes separation from "hard negatives." These hyperparameters directly control diversity in the feature space.

### A.2.2  LATENT-TO-IMAGE SEPARABILITY TRANSFER

Dual Contrastive Inversion applies contrastive learning in both latent and image spaces (Eqs. (13), (14)) . Assume the generator $\theta_G$ is $\mathcal{L}_G$-Lipschitz and the image projection head $h_x$ is $\mathcal{L}_x$-Lipschitz. Then for latents $z_i, z_j$ with corresponding images $x_i = \theta_G(z_i)$ and $x_j = \theta_G(z_j)$,

$$1 - \text{sim}(h_x(x_i), h_x(x_j)) \geq \tfrac{1}{2(\mathcal{L}_G \mathcal{L}_x)^2} \|z_i - z_j\|_2^2. \tag{17}$$

This inequality shows that if $\mathcal{L}_{\text{zcr}}$ enforces latent separation, it automatically guarantees a lower bound on image-level separability for $\mathcal{L}_{\text{scr}}$. Thus latent contrastive learning prevents collapse, while image-level contrast refines visual diversity.

### A.2.3  MODE COVERAGE OF THE cGMM PRIOR

In Distributional Prior Recovery, the cGMM prior (Eq. (5)) models multimodal latent distributions. With $K$ components and smallest weight $\pi_{\min}$, the probability that some mode is missed after $T$ draws is

$$\Pr[\text{missing a mode}] \leq K \exp(-\pi_{\min} T). \tag{18}$$

Hence drawing

$$T \geq \tfrac{1}{\pi_{\min}} \big( \log K + \log(1/\delta) \big) \tag{19}$$

ensures coverage of all $K$ modes with probability $1 - \delta$.

This guarantees that the cGMM prior does not collapse to a single mode. When combined with memory banks in Dual Contrastive Inversion, mode coverage improves further as diverse samples are accumulated over training.

### A.2.4  CONVERGENCE OF DUAL CONTRASTIVE INVERSION UPDATES

Let $\mathcal{J}$ denote the Dual Contrastive Inversion objective (Eq. (12)). The optimization proceeds by alternating updates over latents, the generator, and the projection heads.

$$\mathcal{J}(\theta_G, z, h_x, h_z) = \lambda_1 \mathcal{L}_{\text{scr}} + \lambda_2 \mathcal{L}_{\text{gMI}} + \lambda_3 \mathcal{L}_{\text{zcr}}. \tag{20}$$

**Theorem (Stationarity).**  Under Lipschitz-smoothness assumptions and boundedness below, the updates form a non-increasing sequence. Any limit point $(\theta_G^\star, z^\star, h_x^\star, h_z^\star)$ is stationary:

$$0 \in \partial \mathcal{J}(\theta_G^\star, z^\star, h_x^\star, h_z^\star). \tag{21}$$

Although the problem is non-convex, the training procedure is guaranteed not to diverge. In practice, this explains the empirical stability observed in the experimental Section 4.

### A.2.5  GENERALIZATION BOUNDS VIA FID/JS

We now analyze how well a student trained on synthetic data generalizes to real data. Formally, we compare the expected student loss on the real distribution $P$ and on the synthetic distribution $Q$.

Assume the student loss function $\ell(\cdot)$ is $L$-Lipschitz, meaning that small changes in input samples or their features cannot cause arbitrarily large changes in loss values. Under this assumption, the following inequality holds:

$$\left| \mathbb{E}_{x \sim P} \ell(x) - \mathbb{E}_{x \sim Q} \ell(x) \right| \leq L W_2(P, Q), \tag{22}$$

where $W_2(P, Q)$ is the Wasserstein-2 distance between the real and synthetic distributions.

This bound means that the difference in student performance between training on real data and training on synthetic data is controlled by how close the two distributions are. In practice, when feature distributions are approximated as Gaussians, the Wasserstein-2 distance corresponds to FID, i.e., $W_2^2 = \text{FID}$. Therefore, a smaller FID implies a smaller generalization gap. Similarly, a lower JS divergence indicates better distributional alignment. This shows why improvements in FID/JS reported in our experiments directly translate into more reliable student generalization.

### A.2.6 SUMMARY

In summary, the extended analysis demonstrates that minimizing the InfoNCE loss with sufficiently large memory banks directly maximizes diversity (Eq. (10), Eq. (11)); that latent separability enforced by $\mathcal{L}_{\text{zcr}}$ is guaranteed to transfer to image-level diversity through Lipschitz continuity (Eq. (13), Eq. (14)); that the class-conditional GMM prior ensures coverage of multimodal distributions and prevents collapse (Eq. (5)); that the alternating optimization scheme of Dual Contrastive Inversion converges to stationary points, providing stability (Eq. (12)); and that improvements in FID/JS translate into provably smaller generalization gaps for the student model (Eq. (15)). Together, these results offer a rigorous explanation for why D2CIP consistently enhances both diversity and distillation accuracy.

Unlike prior DFKD methods that depend on fixed or unimodal latent priors, such as random noise initialization or a single Gaussian distribution, D2CIP leverages a learnable class-conditional Gaussian mixture prior that is co-optimized with the inversion process. This design enables the generator to recover class-specific modes and better represent intra-class variability. Furthermore, existing methods such as CMI rely solely on output-level contrastive regularization, which does not explicitly guarantee instance separability or promote sample-level diversity. D2CIP addresses this limitation through dual contrastive learning, in both latent space and sample space, enhanced by memory banks, and theoretically grounded by its connection to diversity maximization objectives. The integration of these components forms a new inversion paradigm that improves mode coverage, increases sample diversity, and enhances downstream distillation effectiveness beyond previous approaches.

### A.3 RESULTS ON IMAGENET-100

To further evaluate the scalability of D2CIP on large-scale datasets, we conduct experiments on ImageNet-100 under the $\mathcal{T}(a)$–$\mathcal{S}(b)$ (ResNet-34 → ResNet-18) setting. As shown in Table 4, many existing DFKD baselines either do not report results or are not reproducible on ImageNet-100 due to its substantial computational overhead. Among the available baselines, CMI achieves 41.04%, PRE-DFKD reaches 43.36%, and TA-DFKD attains 33.66%. In comparison, D2CIP obtains a top-1 accuracy of **51.98%**, substantially outperforming these representative methods. Although the gap to the teacher model (57.80%) is non-trivial, this result demonstrates that D2CIP effectively scales to more challenging large-scale benchmarks, confirming the robustness of distributional priors and dual contrastive learning in improving synthetic data quality and distillation performance.

### A.4 HYPERPARAMETERS SENSITIVITY ANALYSIS

We analyze the sensitivity of critical hyperparameters in the D2CIP framework to better understand their influence on distillation performance. In particular, we vary the weights assigned to different loss components, including distributional alignment, training stability, data transferability, and diversity of synthetic data, and evaluate their effect on the student model's accuracy. Furthermore, we investigate the role of the temperature coefficient in the contrastive loss, which governs the granularity of representation separation. Detailed results are provided in Sections A.4.1 to A.4.5.

Table 4: Performance comparison on ImageNet under the $\mathcal{T}(a)$–$\mathcal{S}(b)$ setting, where Teacher ($\mathcal{T}$) is ResNet-34 and Student ($\mathcal{S}$) is ResNet-18. A dash "–" indicates results either not reported in the original paper or not reproducible under our experimental setting, and **bold** highlights the best performance.

| Method | Top-1 Accuracy (%) |
|---|---|
| Teacher Model ($\mathcal{T}$) | 57.80 |
| Student Model ($\mathcal{S}$) | 57.32 |
| DAFL (ICCV'19) | – |
| ZSKT (ICML'19) | – |
| DFQ (ICCV'19) | – |
| ADI (CVPR'20) | – |
| CMI (IJCAI'21) | 41.04 |
| PRE-DFKD (AAAI'22) | 43.36 |
| TA-DFKD (AAAI'24) | 33.66 |
| **D2CIP (Ours)** | **51.98** |

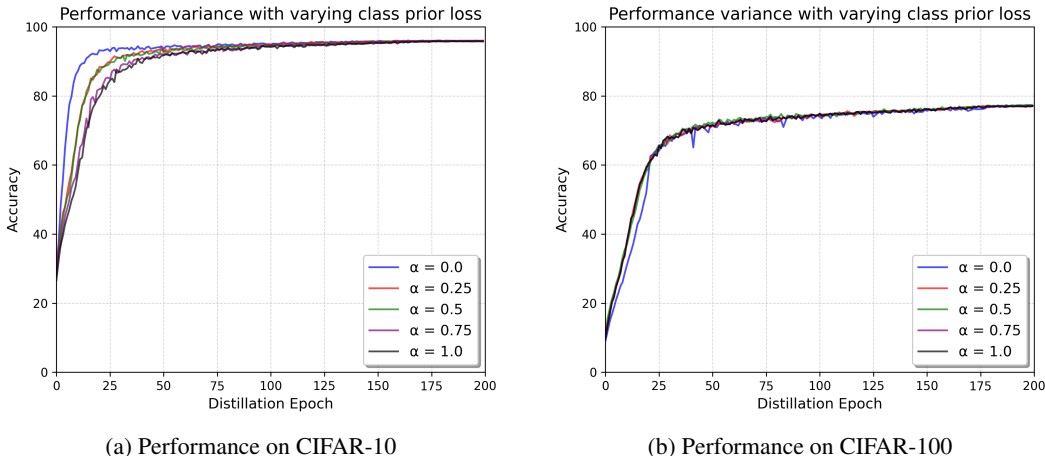

(a) Performance on CIFAR-10      (b) Performance on CIFAR-100

Figure 4: Impact of class prior loss weight $\alpha$ on distillation accuracy across CIFAR-10 and CIFAR-100.

### A.4.1 IMPACT OF DISTRIBUTIONAL SIMILARITY

This subsection focuses on the class prior loss $\mathcal{L}_{\mathrm{cls}}$, which directly embodies the concept of distributional similarity by guiding the synthetic data to match the target class distribution of the original dataset. As a core regularization term, $\mathcal{L}_{\mathrm{cls}}$ enforces alignment between generated and real data distributions, which is fundamental to achieving effective knowledge distillation. To assess its impact, we vary the weighting factor $\alpha$ and conduct experiments on CIFAR-10 and CIFAR-100 with $\alpha$ values set to $\{0.0, 0.25, 0.5, 0.75, 1.0\}$.

As illustrated in Fig. 4, increasing $\alpha$ generally leads to improved distillation accuracy and faster convergence, especially noticeable on the simpler CIFAR-10 dataset. When $\alpha = 0.0$, corresponding to the absence of $\mathcal{L}_{\mathrm{cls}}$, the model exhibits slower convergence and lower final accuracy. In contrast, higher $\alpha$ values promote better class distribution alignment, resulting in more stable and accurate performance. This trend is confirmed by the top-1 accuracy results summarized in Table 5, where the best performance is achieved with $\alpha = 1.0$, reaching **95.16%** on CIFAR-10 and **77.36%** on CIFAR-100.

On the more challenging CIFAR-100 dataset, the effect of increasing $\alpha$ is less pronounced but remains consistent, indicating that while distributional similarity is always beneficial, its impact is stronger when class boundaries are clearer. Overall, these findings confirm that emphasizing the class prior loss improves both convergence and final performance, with larger $\alpha$ values (e.g., 0.75 or 1.0) offering the most reliable gains.

Table 5: Top-1 Accuracy with different class prior loss weights $\alpha$ on CIFAR-10 and CIFAR-100.

| Dataset | varying $\alpha$ | | | | |
|---|---|---|---|---|---|
| | 0.0 | 0.25 | 0.5 | 0.75 | 1.0 |
| CIFAR-10 | 94.96 | 95.01 | 94.91 | 95.00 | 95.16 |
| CIFAR-100 | 76.35 | 77.03 | 77.20 | 77.07 | 77.36 |

These results highlight that emphasizing the class prior loss is crucial for enforcing distributional alignment, leading to faster convergence and stronger distillation performance in D2CIP.

### A.4.2 IMPACT OF TRAINING STABILITY

This subsection focuses on the batch normalization loss $\mathcal{L}_{bn}$, which plays a critical role in enhancing training stability by aligning the feature statistics of synthetic data with those of the target model. As a regularization term, $\mathcal{L}_{bn}$ mitigates distributional shifts during generator optimization, promoting smoother convergence and improved distillation performance. To assess its impact, we vary the weighting factor $\beta$ and conduct experiments on CIFAR-10 and CIFAR-100 with $\beta$ values set to {0.0, 0.25, 0.5, 0.75, 1.0}.

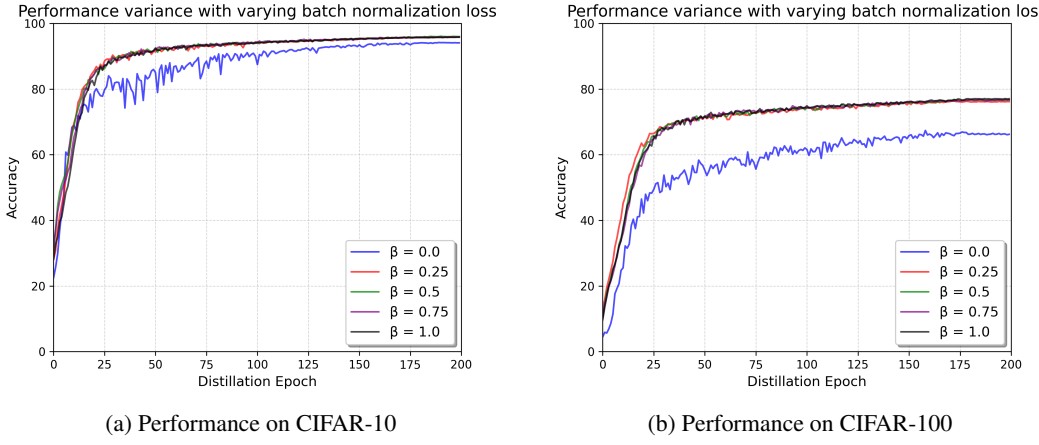

(a) Performance on CIFAR-10       (b) Performance on CIFAR-100

Figure 5: Impact of class prior loss weight $\beta$ on distillation accuracy across CIFAR-10 and CIFAR-100.

As illustrated in Fig. 5, increasing $\beta$ leads to notable improvements in both convergence speed and final accuracy, particularly on the more challenging CIFAR-100 dataset. Without $\mathcal{L}_{bn}$ ($\beta = 0.0$), the model suffers from unstable training and significantly lower accuracy. Introducing the batch normalization loss helps stabilize feature distributions, resulting in higher and more stable performance. This trend is further confirmed by the results in Table 6, where the best performance is achieved at $\beta = 0.5$ for CIFAR-10 with **95.16%** accuracy, and at $\beta = 1.0$ for CIFAR-100 with **77.36%** accuracy.

Table 6: Top-1 Accuracy with different batch normalization loss weights $\beta$ on CIFAR-10 and CIFAR-100.

| Dataset | varying $\beta$ | | | | |
|---|---|---|---|---|---|
| | 0.0 | 0.25 | 0.5 | 0.75 | 1.0 |
| CIFAR-10 | 93.18 | 94.87 | 95.16 | 94.85 | 94.83 |
| CIFAR-100 | 66.92 | 76.25 | 76.69 | 76.80 | 77.36 |

In summary, incorporating $\mathcal{L}_{\text{bn}}$ effectively enhances the stability of generator training and improves the quality of synthetic data. Assigning moderate to high values of $\beta$ (e.g., 0.5 or 1.0) yields the most consistent accuracy gains across datasets.

### A.4.3 IMPACT OF DATA TRANSFERABILITY

This subsection investigates the role of adversarial distillation loss $\mathcal{L}_{\text{adv}}$, which directly improves data transferability by encouraging the generator to produce challenging synthetic samples near the decision boundary of the teacher model. This process enhances the student model's generalization by exposing it to more informative and transferable synthetic data during distillation. To evaluate the impact of $\mathcal{L}_{\text{adv}}$, we vary its weighting factor $\gamma$ and conduct experiments on CIFAR-10 and CIFAR-100 with $\gamma$ values set to {0.0, 0.25, 0.5, 0.75, 1.0}.

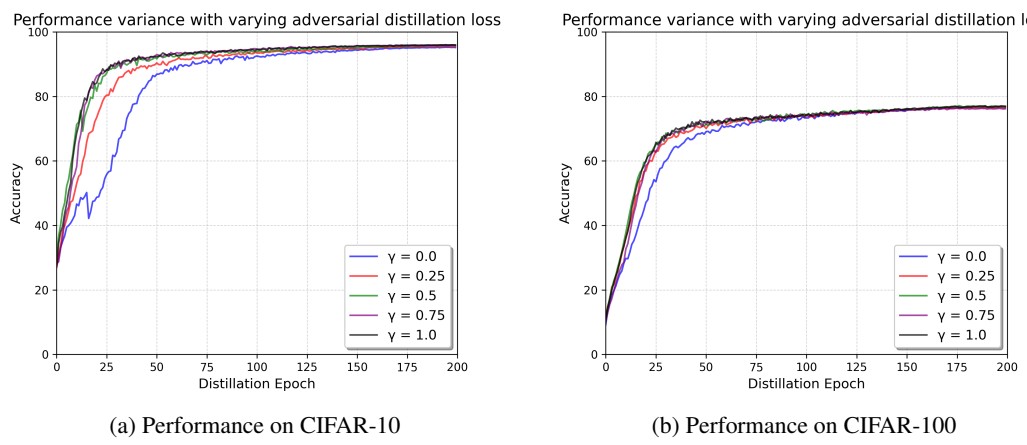

(a) Performance on CIFAR-10        (b) Performance on CIFAR-100

Figure 6: Impact of adversarial distillation loss weight $\gamma$ on distillation accuracy across CIFAR-10 and CIFAR-100.

As shown in Fig. 6, increasing $\gamma$ generally leads to better distillation performance and faster convergence, particularly evident on the CIFAR-10 dataset. Without $\mathcal{L}_{\text{adv}}$ ($\gamma = 0.0$), the model exhibits slower convergence and lower accuracy, indicating that adversarial distillation effectively guides the student model to learn more discriminative features. Table 7 further confirms this trend, where the best performance is achieved at $\gamma = 0.5$, reaching **95.16%** on CIFAR-10 and **77.36%** on CIFAR-100.

Table 7: Top-1 Accuracy with different adversarial distillation loss weights $\gamma$ on CIFAR-10 and CIFAR-100.

| Dataset | varying $\gamma$ | | | | |
|---|---|---|---|---|---|
| | 0.0 | 0.25 | 0.5 | 0.75 | 1.0 |
| CIFAR-10 | 94.27 | 94.58 | **95.16** | 94.99 | 95.01 |
| CIFAR-100 | 76.76 | 76.58 | **77.36** | 76.29 | 77.07 |

While the performance improvements on CIFAR-100 are relatively moderate, the consistent gains demonstrate that incorporating adversarial distillation improves data transferability across different tasks. Assigning moderate values to $\gamma$ (e.g., 0.5) achieves a good balance between training stability and performance.

### A.4.4 IMPACT OF SYNTHETIC DATA DIVERSITY

This subsection analyzes the effect of the synthetic contrastive loss $\mathcal{L}_{\text{scr}}$, which directly enhances the diversity of generated synthetic data by encouraging instance-level discrimination. This diversity is critical for producing high-quality synthetic datasets that capture rich variations within each class,

leading to better generalization in the student model. To evaluate its impact, we vary the weighting factor $\lambda_1$ across $\{0.0, 0.2, 0.4, 0.6, 0.8\}$ and measure performance on CIFAR-10 and CIFAR-100.

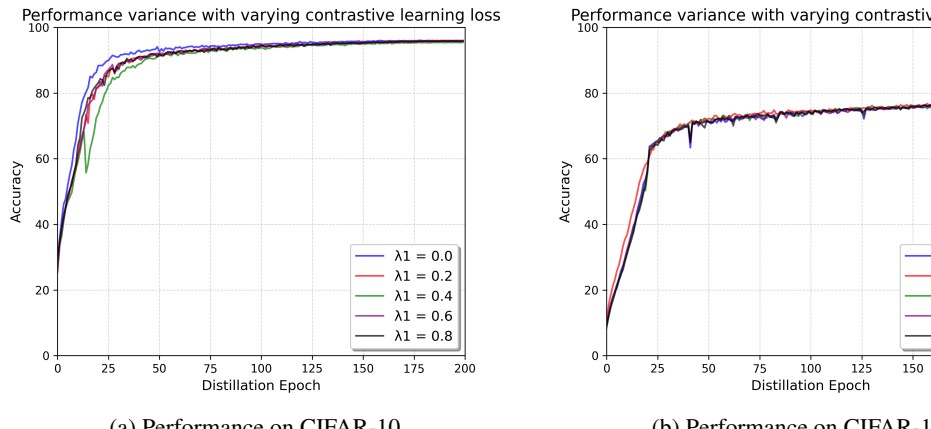

(a) Performance on CIFAR-10        (b) Performance on CIFAR-100

Figure 7: Impact of synthetic contrastive loss weight $\lambda_1$ on distillation accuracy across CIFAR-10 and CIFAR-100.

As illustrated in Fig. 7, introducing $\mathcal{L}_{\text{scr}}$ ($\lambda_1 > 0$) improves both the convergence speed and final distillation accuracy, particularly noticeable on CIFAR-100. When $\lambda_1 = 0.0$, indicating the absence of diversity constraints, the model shows limited improvement and lower final accuracy. As $\lambda_1$ increases, the generator produces more diverse and informative synthetic samples, which facilitates better knowledge transfer to the student model. This is further confirmed by the top-1 accuracy results in Table 8, where the highest performance is achieved at $\lambda_1 = 0.8$, reaching **95.16%** on CIFAR-10 and **77.36%** on CIFAR-100.

Table 8: Top-1 Accuracy with different synthetic contrastive loss weights $\lambda_1$ on CIFAR-10 and CIFAR-100.

| Dataset | varying $\lambda_1$ | | | | |
|---|---|---|---|---|---|
| | 0.0 | 0.2 | 0.4 | 0.6 | 0.8 |
| CIFAR-10 | 94.44 | 95.00 | 94.45 | 94.89 | 95.16 |
| CIFAR-100 | 76.23 | 76.03 | 76.38 | 77.02 | 77.36 |

However, extremely large values of $\lambda_1$ may lead to diminishing returns, as overly focusing on diversity could introduce noisy or less representative samples. These results suggest that assigning a moderate to high $\lambda_1$ value (e.g., 0.6 or 0.8) effectively improves synthetic data diversity without sacrificing sample quality.

### A.4.5    IMPACT OF TEMPERATURE COEFFICIENCY IN CONTRASTIVE LOSS

This subsection investigates the influence of the temperature coefficient $\tau$ in the contrastive loss, which plays a key role in controlling the sharpness of similarity distributions during representation learning. A well-chosen $\tau$ can effectively balance the hardness of negative samples and improve representation discrimination, thereby enhancing the quality of synthetic data for knowledge distillation. To analyze its impact, we vary $\tau$ across $\{0.0, 0.1, 0.2, 0.3, 0.4\}$ and evaluate the performance on CIFAR-10 and CIFAR-100.

As shown in Fig. 8, increasing $\tau$ generally leads to significant improvements in distillation accuracy and faster convergence. When $\tau = 0.0$, which effectively disables the contrastive loss, the model exhibits slow convergence and poor final accuracy, particularly on CIFAR-100. As $\tau$ increases, the model learns more informative feature representations, resulting in stronger distillation performance.

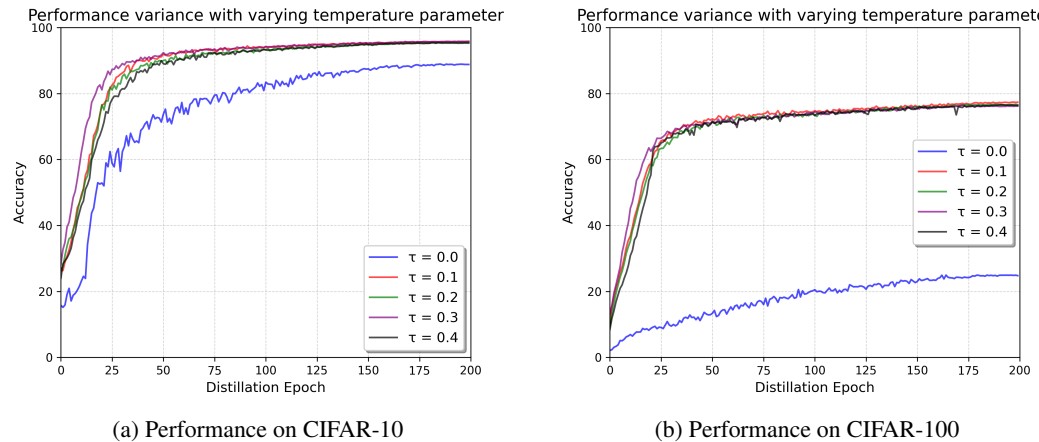

(a) Performance on CIFAR-10          (b) Performance on CIFAR-100

Figure 8: Impact of temperature coefficiency $\tau$ in contrastive loss on distillation accuracy across CIFAR-10 and CIFAR-100.

This trend is further confirmed by the top-1 accuracy results in Table 9, where the best accuracy is achieved at $\tau = 0.25$, reaching **95.16%** on CIFAR-10 and **77.36%** on CIFAR-100.

Table 9: Top-1 Accuracy with different temperature coefficiency $\tau$ in contrastive loss on CIFAR-10 and CIFAR-100.

| Dataset | varying $\tau$ | | | | |
|---|---|---|---|---|---|
| | 0.0 | 0.1 | 0.2 | 0.3 | 0.4 |
| CIFAR-10 | 87.89 | 95.16 | 94.51 | 94.83 | 94.39 |
| CIFAR-100 | 24.92 | 77.36 | 76.71 | 76.22 | 76.59 |

However, setting $\tau$ excessively high does not bring further benefits and may slightly degrade performance, likely due to oversmoothing the similarity distribution and reducing contrastive learning effectiveness. These results suggest that a moderate $\tau$ value (e.g., 0.25) provides the optimal trade-off between representation discrimination and training stability across datasets.

Table 10: Empirical comparison of GPU memory usage and per-epoch runtime for CMI, TA-DFKD, and D2CIP.

| Method | GPU Memory (MB) | Epoch Time |
|---|---|---|
| CMI | 2244.53 | 7m 27s |
| TA-DFKD | 3267.52 | 7m 43s |
| D2CIP | 3779.79 | 8m 12s |

### A.5 QUANTITATIVE EVALUATION OF SYNTHETIC DATA

To comprehensively assess the quality and diversity of the generated synthetic data, we conduct a quantitative evaluation using both established and supplementary metrics. These evaluations aim to measure how closely the synthetic data distribution aligns with the real data distribution, providing insights into the effectiveness of the generation process. Specifically, we focus on evaluating the class-wise distribution consistency on the CIFAR-10 dataset, which serves as a representative benchmark for visual data generation tasks.

Section A.5.1 introduces the evaluation metrics, and Section A.5.2 presents the corresponding quantitative results.

### A.5.1 DEFINITIONS OF DIVERSITY MEASURE METRICS

In this subsection, we formally define the two primary metrics used to evaluate the quality and diversity of generated data: Fréchet Inception Distance (FID) and Jensen-Shannon (JS) divergence. In addition, we introduce two supplementary metrics to measure the distributional discrepancy between real data $\mathcal{X}$ and generated data $\mathcal{X}'$.

**Fréchet Inception Distance (FID)**    FID measures the distance between feature distributions of $\mathcal{X}$ and $\mathcal{X}'$, typically extracted using a pretrained Inception network. Assuming that the features follow multivariate Gaussian distributions, FID is computed as:

$$\text{FID}(\mathcal{X}, \mathcal{X}') = \|\mu_{\mathcal{X}} - \mu_{\mathcal{X}'}\|_2^2 + \text{Tr}\left(\Sigma_{\mathcal{X}} + \Sigma_{\mathcal{X}'} - 2\left(\Sigma_{\mathcal{X}}\Sigma_{\mathcal{X}'}\right)^{1/2}\right) \tag{23}$$

where $\mu_{\mathcal{X}}, \Sigma_{\mathcal{X}}$ and $\mu_{\mathcal{X}'}, \Sigma_{\mathcal{X}'}$ are the means and covariances of real and generated feature distributions, respectively. A lower FID indicates better quality and diversity of generated data Heusel et al. (2017).

**Jensen-Shannon (JS) Divergence**    JS divergence is a symmetric and bounded measure of similarity between two distributions $P$ and $Q$, corresponding to $\mathcal{X}$ and $\mathcal{X}'$. It is defined as:

$$\text{JS}(P\|Q) = \frac{1}{2}\text{KL}\left(P\|M\right) + \frac{1}{2}\text{KL}\left(Q\|M\right), \quad \text{where } M = \frac{1}{2}(P + Q) \tag{24}$$

Here, $\text{KL}(\cdot\|\cdot)$ denotes the Kullback-Leibler divergence. JS divergence ranges from 0 to 1, with lower values indicating higher similarity Lin (2002).

**Maximum Mean Discrepancy (MMD)**    MMD directly compares two distributions using kernel embeddings:

$$\text{MMD}^2(\mathcal{X}, \mathcal{X}') = \mathbb{E}_{x,x'\sim\mathcal{X}}[k(x, x')] + \mathbb{E}_{\tilde{x},\tilde{x}'\sim\mathcal{X}'}[k(\tilde{x}, \tilde{x}')] - 2\mathbb{E}_{x\sim\mathcal{X},\tilde{x}\sim\mathcal{X}'}[k(x, \tilde{x})] \tag{25}$$

where $k(\cdot, \cdot)$ is a characteristic kernel function, typically a Gaussian kernel. Lower MMD values imply closer alignment between the two distributions Gretton et al. (2012).

**Learned Perceptual Image Patch Similarity (LPIPS)**    The Learned Perceptual Image Patch Similarity (LPIPS) metric Zhang et al. (2018) is a deep learning-based perceptual similarity measure, which has been shown to better correlate with human visual perception compared to traditional metrics such as Mean Squared Error (MSE) and Structural Similarity Index Measure (SSIM).

Unlike pixel-wise similarity metrics, LPIPS evaluates the perceptual similarity between two data instances, denoted as $\mathcal{X}$ and $\mathcal{X}'$, by comparing their deep feature representations extracted from a pre-trained convolutional neural network, such as AlexNet or VGG. This approach enables the similarity evaluation to capture high-level semantic information rather than low-level pixel differences.

Formally, the LPIPS score between $\mathcal{X}$ and $\mathcal{X}'$ is computed as:

$$\text{LPIPS}(\mathcal{X}, \mathcal{X}') = \sum_l \frac{1}{H_l W_l} \sum_{h=1}^{H_l} \sum_{w=1}^{W_l} \|w_l \odot (f_l(\mathcal{X})_{h,w} - f_l(\mathcal{X}')_{h,w})\|_2^2 \tag{26}$$

where:

- $f_l(\cdot)$ denotes the feature maps extracted from the $l$-th layer of the pre-trained network.
- $H_l$ and $W_l$ are the spatial dimensions of the feature maps at layer $l$.
- $w_l$ represents learned channel-wise weights that adaptively modulate the importance of feature differences.
- $\odot$ denotes element-wise multiplication.

By operating in the deep feature space, LPIPS effectively captures both low-level texture and high-level semantic differences, making it more robust for evaluating the visual fidelity and perceptual quality of generated or reconstructed data. This property is particularly important in generative modeling scenarios, where perceptual realism is often more critical than precise pixel-wise accuracy.

### A.5.2 ADDITIONAL QUANTITATIVE RESULTS FOR SYNTHETIC DATA DIVERSITY

In addition to the primary evaluation metrics, we further assess the diversity of the generated data using two complementary measures: Maximum Mean Discrepancy (MMD) and Learned Perceptual Image Patch Similarity (LPIPS). These metrics provide additional insights into the distributional similarity between real data $\mathcal{X}$ and generated data $\mathcal{X}'$, capturing both statistical distribution alignment and perceptual similarity.

The summarized results are presented in Table 11. Our proposed method (D2CIP) consistently achieves the lowest MMD and LPIPS values across the majority of evaluated classes on CIFAR-10. Specifically, for MMD, D2CIP obtains the lowest average score of 0.279, which outperforms both the CMI (0.286) and TA-DFKD (0.439) baselines. This suggests that the synthetic data generated by D2CIP better matches the statistical properties of the real data distribution.

In terms of perceptual similarity, as measured by LPIPS, D2CIP also achieves the lowest average score of 0.183, compared to 0.201 for CMI and 0.189 for TA-DFKD. Notably, D2CIP outperforms the other methods in most object categories, such as *airplane* (0.157), *automobile* (0.182), *cat* (0.184), and *horse* (0.167), indicating that the generated samples not only align better statistically but also possess higher visual fidelity and diversity from a perceptual perspective.

Table 11: Quantitative comparison of data generation diversity among CMI, TA-DFKD, and D2CIP using **MMD** and **LPIPS** across classes on CIFAR-10. The symbol ↓ indicates that lower values are preferred.

| Class | MMD (↓) | | | LPIPS (↓) | | |
| --- | --- | --- | --- | --- | --- | --- |
| | **CMI** | **TA-DFKD** | **D2CIP** | **CMI** | **TA-DFKD** | **D2CIP** |
| airplane | **0.313** | 0.434 | 0.353 | 0.219 | 0.193 | **0.157** |
| automobile | **0.263** | 0.361 | 0.286 | 0.192 | 0.189 | **0.182** |
| bird | 0.258 | 0.459 | **0.188** | 0.215 | **0.189** | 0.190 |
| cat | 0.265 | 0.427 | **0.210** | 0.205 | 0.188 | **0.184** |
| deer | 0.301 | 0.509 | **0.270** | 0.200 | 0.195 | **0.189** |
| dog | **0.248** | 0.441 | 0.331 | 0.195 | **0.187** | 0.190 |
| frog | 0.341 | 0.551 | **0.274** | 0.221 | 0.190 | **0.177** |
| horse | 0.280 | 0.455 | **0.231** | 0.182 | 0.185 | **0.167** |
| ship | **0.299** | 0.399 | 0.306 | 0.204 | **0.187** | 0.208 |
| truck | **0.292** | 0.354 | 0.344 | **0.185** | 0.186 | 0.187 |
| Average | 0.286 | 0.439 | **0.279** | 0.201 | 0.189 | **0.183** |

It is worth highlighting that certain object classes such as *ship* and *truck* exhibit marginal differences between methods, suggesting that these categories might be less sensitive to the diversity enhancements introduced by D2CIP. However, the overall lower average values demonstrate the robustness of D2CIP across diverse classes.

These findings are consistent with earlier results based on JS divergence and FID (see Fig. 2), further validating the effectiveness of our method in producing high-quality and diverse synthetic data. The joint improvements in both statistical and perceptual diversity suggest that D2CIP-generated samples are not only more distinguishable and varied but also better aligned with human perceptual expectations.

Overall, these results demonstrate that D2CIP effectively enhances the diversity and quality of synthetic data, which is critical for improving the generalization performance of downstream models trained on such data. By achieving lower values on both MMD and LPIPS metrics, D2CIP provides a more balanced and comprehensive solution for realistic synthetic data generation.

### A.5.3 ADDITIONAL VISUALIZATION RESULTS FOR SYNTHETIC DATA DIVERSITY

To provide intuitive insights into the distributional differences between real and synthetic data, we visualize the feature embeddings using a two-dimensional projection obtained from a well-trained ResNet34 model on CIFAR-10. As shown in Figures 9 and 10, the comparisons illustrate how different generation methods affect the distribution alignment across various object classes.

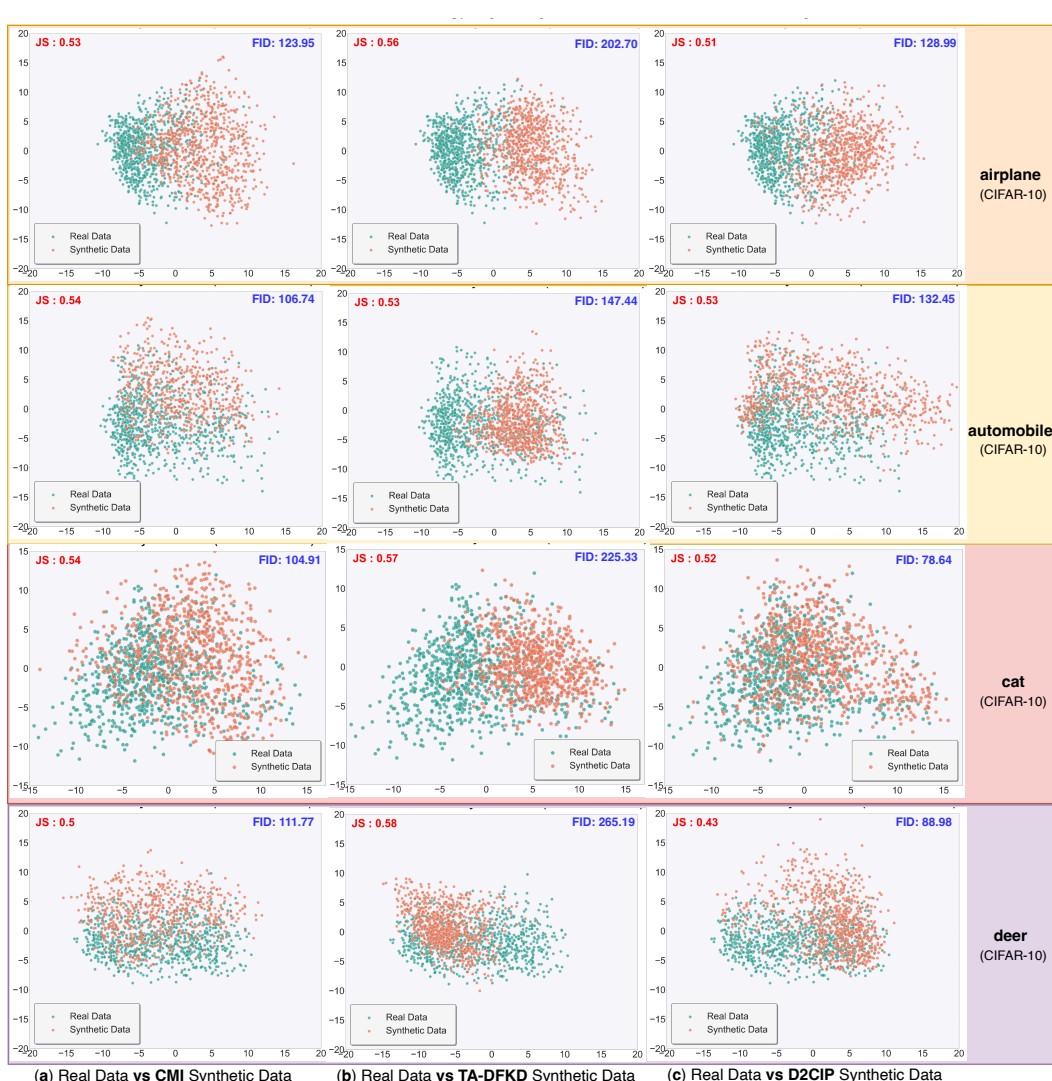

(a) Real Data **vs CMI** Synthetic Data    (b) Real Data **vs TA-DFKD** Synthetic Data    (c) Real Data **vs D2CIP** Synthetic Data

Figure 9: Distribution visualization of real and synthetic data using a two-dimensional mapping from a well-trained ResNet34 on CIFAR-10. (**a**) Real vs CMI-generated data for airplane, automobile, cat, and deer from top to bottom. (**b**) Real vs TA-DFKD-generated data for airplane, automobile, cat, and deer from top to bottom. (**c**) Real vs D2CIP-generated data for airplane, automobile, cat, and deer from top to bottom.

### A.6 ALGORITHMIC COMPUTATIONAL COMPLEXITY ANALYSIS

In this section, we provide a comprehensive analysis of the computational complexity of the proposed D2CIP framework, along with a discussion on the underlying design rationale and its performance trade-offs. While the algorithm introduces non-negligible computational costs, these are well-justified by the substantial improvements in data diversity, model robustness, and generalization capabilities.

**Notation Overview**    The key factors influencing computational complexity are summarized as follows:

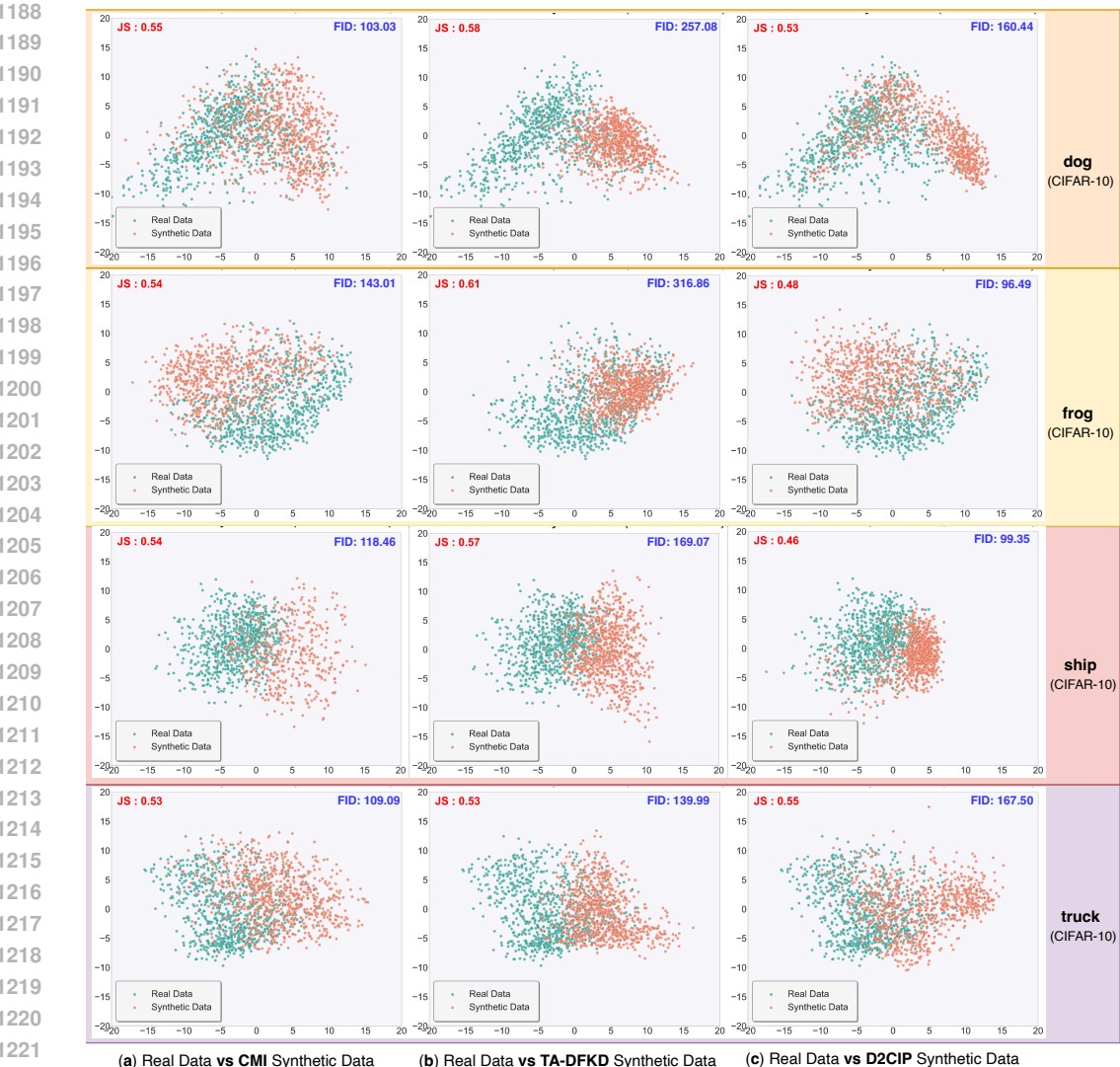

Figure 10: Distribution visualization of real and synthetic data using a two-dimensional mapping from a well-trained ResNet34 on CIFAR-10. (a) Real vs CMI-generated data for dog, frog, ship, and truck from top to bottom. (b) Real vs TA-DFKD-generated data for dog, frog, ship, and truck from top to bottom. (c) Real vs D2CIP-generated data for dog, frog, ship, and truck from top to bottom.

- $C$: Number of target classes.
- $K$: Number of Gaussian components in the learned cGMM prior.
- $d$: Latent space dimensionality.
- $B$: Batch size.
- $C$: Number of target classes.
- $f_G$, $f_T$: Computational costs of the generator $\theta_G$ and the target model $\theta_T$.
- $f_{h_x}$, $f_{h_z}$: Computational costs of the embedding heads $h_x$ and $h_z$.

**Complexity of Distributional Prior Recovery**   In this stage, the cGMM prior is estimated by aligning the latent distribution with teacher predictions and batch-normalization statistics. The complexity sources are:

- Sampling from the cGMM prior requires drawing a mixture component for each class, with complexity $\mathcal{O}(C \cdot K \cdot d)$.

- Forward passes through generator and teacher models add $\mathcal{O}(f_G + f_T)$.
- Distributional alignment using class priors contributes $\mathcal{O}(B \cdot C)$, comparable to standard classification losses.

Thus, the total per-iteration complexity is:

$$\mathcal{O}\left(C \cdot K \cdot d + f_G + f_T + B \cdot C\right).$$

Although more costly than a unimodal Gaussian prior, the cGMM design provides fine-grained modeling of multimodal class distributions, which is critical for generating class-consistent synthetic data.

**Complexity of Dual Contrastive Inversion** This stage enhances both latent and instance-level diversity through dual contrastive learning:

- The computation of latent contrastive loss $\mathcal{L}_{zcr}$ and synthetic contrastive loss $\mathcal{L}_{scr}$ requires pairwise similarity comparisons within each batch, yielding a complexity of $\mathcal{O}(B^2 \cdot d)$ for each. This quadratic cost dominates the stage, but it is a deliberate design choice that enforces fine-grained discrimination in both latent and data spaces, effectively mitigating mode collapse.
- The use of memory banks $\mathcal{B}_z$ and $\mathcal{B}_x$ introduces additional storage and indexing operations. However, these operations grow only linearly with the history size and are negligible compared to pairwise similarity computations, while substantially enriching negative sample diversity.
- Forward and backward passes through the generator $\theta_G$ and embedding heads $h_x$, $h_z$ contribute $\mathcal{O}(f_G + f_{h_x} + f_{h_z})$, which are standard neural network costs and scale with model size. These updates are essential for refining both sample quality and the discriminative power of latent embeddings.
- Teacher model inference adds $\mathcal{O}(f_T)$ per iteration. Although this is an additional cost, it is necessary to enforce alignment with the target distribution and ensure meaningful supervision during synthetic data generation.

The total complexity per iteration in this phase is:

$$\mathcal{O}\left(K \cdot d + 2B^2 \cdot d + f_G + f_T + f_{h_x} + f_{h_z}\right).$$

**Empirical Runtime and Memory Analysis** To complement the theoretical complexity analysis, we further provide an empirical study of runtime and memory overhead compared to baseline methods. Specifically, we benchmark GPU memory usage and per-epoch training time of D2CIP against two representative baselines (CMI and TA-DFKD). As summarized in Table 10, D2CIP requires moderately higher resources (e.g., 3779.79 MB and 8m 12s per epoch) compared to CMI (2244.53 MB and 7m 27s per epoch) and TA-DFKD (3267.52 MB and 7m 43s per epoch).

Beyond runtime and memory considerations, we further examine the training dynamics of the three methods. As shown in Figure 11, D2CIP does not exhibit the fastest initial drop in loss; however, its convergence trajectory remains consistently lower and more stable than those of TA-DFKD and PRE-DFKD throughout training. TA-DFKD descends rapidly in the early stages but plateaus at a noticeably higher loss level and shows larger fluctuations across epochs. PRE-DFKD converges more slowly and maintains the highest overall loss. In contrast, D2CIP continues to improve steadily and ultimately achieves the **lowest final training loss** among all methods. These results indicate that although D2CIP emphasizes stability rather than aggressive early descent, it provides **superior final optimization quality** and avoids the stagnation behavior observed in the baselines.

These results confirm that, while D2CIP introduces additional overhead due to its class-conditional prior and dual contrastive objectives, the increase remains tractable and consistent across datasets owing to the use of a fixed hyperparameter configuration. More importantly, this trade-off is well justified by the substantial gains in both diversity and distillation performance (see Figures 2–3 and Table 1). This demonstrates that the added computational cost directly translates into measurable performance improvements.

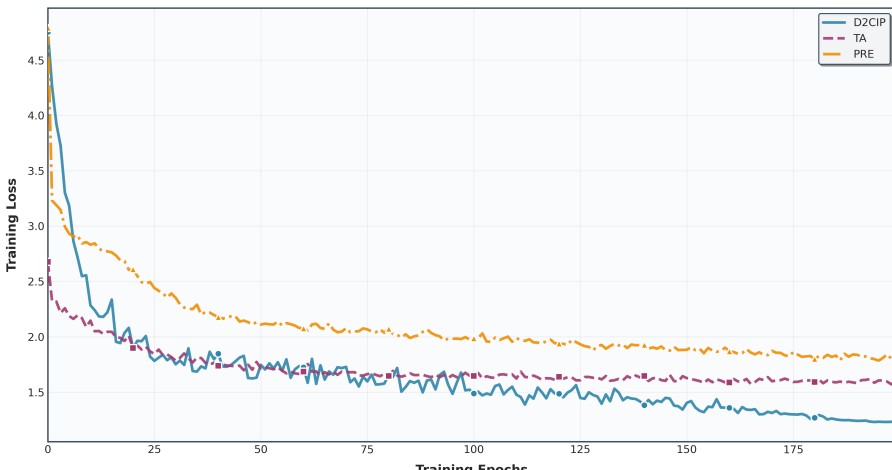

Figure 11: Training loss curves of D2CIP and two baselines (TA-DFKD and PRE-DFKD) on CIFAR-100.

**Rationale for Computational Design Choices**   While the quadratic dependence on batch size in contrastive loss calculations introduces significant overhead, it is important to recognize that this design explicitly targets improved data diversity and the mitigation of mode collapse—issues that are particularly critical in generative modeling. By enforcing a stronger separation between synthetic instances in the embedding space, the framework fosters better generalization and downstream task performance.

Moreover, this increased complexity is a one-time cost incurred during the offline synthetic data generation and model inversion process. Once the synthetic data bank is established, subsequent model training (e.g., knowledge distillation or classifier training) benefits from significantly reduced data acquisition costs, improved model robustness, and enhanced resistance to adversarial perturbations.

This trade-off between computational cost and performance gain is particularly favorable in scenarios where data collection is expensive or privacy constraints limit access to real data, such as in healthcare or financial applications.

**Balanced Trade-offs and Justifications**   Although the proposed framework introduces additional computational overhead, this design choice is motivated by the substantial improvements it brings in terms of data diversity and model generalization. The incorporation of contrastive learning, despite its quadratic complexity with respect to batch size, is not merely a computational burden but a critical mechanism for addressing long-standing challenges such as mode collapse and poor sample diversity in generative models. By explicitly encouraging instance-level discriminability in both the latent and synthetic data spaces, the framework ensures that generated samples are more diverse and informative, ultimately leading to models that generalize better to unseen data.

It is also important to note that the majority of this computational cost is incurred during the offline synthetic data generation phase. Once the synthetic dataset is constructed, subsequent training and deployment phases can proceed efficiently, benefiting from a richer and more representative training corpus without the need for repeated access to real, often sensitive, data. This makes the framework particularly attractive for applications in privacy-sensitive domains such as healthcare and finance, where acquiring and sharing real data is both costly and challenging.

Moreover, although the theoretical complexity appears high, practical implementation benefits from modern hardware acceleration and algorithmic optimizations. Pairwise similarity computations, which dominate the complexity, can be efficiently parallelized across GPUs or distributed systems, significantly reducing wall-clock training time. Techniques such as memory-efficient contrastive learning and approximate similarity calculations can further mitigate runtime costs without sacrificing performance.

In summary, the computational complexity of the proposed method should be viewed not as a limitation but as a deliberate investment toward achieving higher-quality synthetic data and stronger

downstream performance. This trade-off is well-justified by the observed improvements in both robustness and generalization, making the framework a practical and effective solution for scenarios where data diversity and privacy preservation are paramount.

**Conclusion** The proposed D2CIP framework strategically balances computational complexity with significant performance improvements in data diversity, model robustness, and generalization. The higher computational demands are carefully justified by their direct contribution to solving fundamental challenges in synthetic data generation. When viewed from a system-level perspective, the trade-offs are not only reasonable but also highly beneficial for real-world deployment scenarios where high-quality, diverse data is often scarce or difficult to obtain.

### A.7 VISUALIZATION OF SYNTHETIC DATA

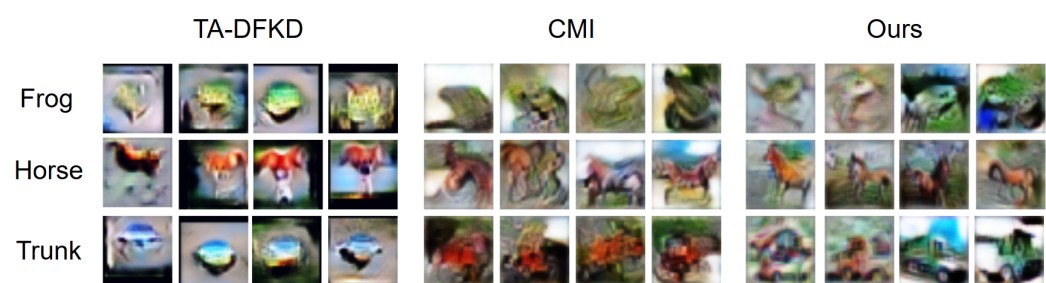

Figure 12: Inverted data from a pre-trained ResNet-34 on CIFAR-10. While TA-DFKD generates images by clustering high-quality synthetic samples via cGMM and CMI through contrastive learning, our approach achieves superior visual quality and diversity.

In addition to quantitative evaluations, we provide visual comparisons of the generated data to offer more intuitive insights into the visual quality and diversity of different methods. Visual inspection serves as a complementary tool to numerical metrics, allowing us to assess whether the generated samples exhibit realistic semantic features and class-relevant patterns.

Figure 12 presents the inverted data generated from a pre-trained ResNet-34 model on the CIFAR-10 dataset. Three representative classes (*Frog*, *Horse*, and *Truck*) are selected to showcase the comparative results of three methods: TA-DFKD, CMI, and our proposed approach. Each row in the figure corresponds to a specific class, and each column presents the generated samples from the respective method.

From the visualization, several key observations can be made:

- **TA-DFKD:** The samples generated by TA-DFKD often suffer from significant visual artifacts and lack clear semantic structures. Although some recognizable shapes appear, the overall image quality is poor, and class-specific features are not well preserved. This reflects the limitations of TA-DFKD in maintaining both visual fidelity and class consistency.

- **CMI:** The CMI method generates samples with moderately improved visual clarity compared to TA-DFKD. Some class-relevant features, such as rough shapes of frogs and horses, begin to emerge. However, the generated images still appear blurry, and fine-grained details are missing. This indicates that while contrastive learning improves sample quality, it struggles to fully capture complex visual patterns.

- **Ours:** In contrast, our proposed method produces visually realistic and semantically meaningful samples. The generated images clearly exhibit recognizable shapes and textures associated with each class. For example, the frog samples display distinct amphibian features, the horse images reveal clear body shapes and posture, and the truck samples capture recognizable structural patterns. This demonstrates the effectiveness of our approach in generating high-fidelity and diverse samples that align closely with real-world object appearances.

These visual results align with our quantitative findings presented in Section A.5.2, confirming that our method consistently outperforms baseline approaches. By effectively integrating contrastive learning and advanced distribution alignment techniques, our approach ensures not only numerical improvements but also significant enhancements in visual realism and semantic alignment.

## A.8    IMPLEMENTATION DETAILS

All experiments were conducted using the PyTorch framework on an NVIDIA GeForce RTX 4090 GPU. The proposed D2CIP method adopts a two-stage training scheme: *Distributional Recovery* and *Gaussian Mixture Distributional Model Inversion (D2CIP)*.

**Stage 1: Distributional Prior Recovery.** In this stage, the generator $\theta_G$ and the Gaussian Mixture Model prior $\mathcal{P}_{\mathrm{gmm}}$ are jointly optimized to learn a rich latent distribution for diverse data generation. The objective function combines the class prior loss $\mathcal{L}_{\mathrm{cls}}$ and the batch normalization regularization loss $\mathcal{L}_{\mathrm{bn}}$, balanced by hyperparameters $\alpha$ and $\beta$. The cGMM prior is trained for 10,000 iterations using the reparameterization trick for latent sampling.

**Stage 2: Dual Contrastive Inversion.** In this stage, $\theta_G$, latent codes $z$, and embedding headers $h_z$ and $h_x$ are jointly optimized to enhance both latent and instance-level diversity through dual contrastive learning. The total loss includes the instance-level contrastive loss $\mathcal{L}_{\mathrm{scr}}$, the latent-level contrastive loss $\mathcal{L}_{\mathrm{zcr}}$, and the Gaussian Mixture Inversion loss $\mathcal{L}_{\mathrm{gMI}}$, weighted by $\lambda_1$, $\lambda_2$, and $\lambda_3$, respectively.

In each training cycle, 300 batches of 256 synthetic samples are generated, and the most informative batch is selected using the decision adversarial loss $\mathcal{L}_{\mathrm{adv}}$ to guide knowledge distillation.

Table 12: Hyperparameter Settings

| Category | Parameter (Symbol) | Value |
|---|---|---|
| D2CIP: Distributional Prior Recovery | Class Prior Loss Weight ($\alpha$) | 1.0 |
| | BatchNorm Loss Weight ($\beta$) | 1.0 |
| | cGMM Training Iterations | 10,000 |
| | Number of cGMM Components ($K$) | Variable |
| D2CIP: Dual Contrastive Inversion | Contrastive Loss Weight ($\lambda_1$) | 0.8 |
| | gMI Loss Weight ($\lambda_2$) | 1.0 |
| | Latent Contrastive Loss Weight ($\lambda_3$) | 1.0 |
| | Temperature for Contrastive Loss ($T$) | 0.1 |
| | Number of Cycles | 5 |
| D2CIP-based DFKD | Adversarial Loss Weight ($\gamma$) | 0.5 |
| | Batch Size | 256 |
| | Batches per Cycle | 300 |

This hyperparameter configuration ensures a balanced contribution from each loss component, enabling the generation of diverse and high-quality synthetic data for effective knowledge distillation.

## A.9    DETAILED RELATED WORKS

**Model Inversion (MI) for Data-Free Knowledge Distillation (DFKD)** Model inversion (MI) techniques, traditionally raising privacy concerns, have been repurposed for data-free knowledge distillation (DFKD) by synthesizing training data. Fang et al. Fang et al. (2021c) proposed Contrastive Model Inversion (CMI) to mitigate mode collapse by enhancing data diversity. Binici et al. Binici et al. (2022) improved knowledge distillation using Condensed Sample-Guided Model Inversion. Kang et al. Kang et al. (2023) introduced FRAMI for generating high-quality Mel-spectrograms, while Liu et al. Liu et al. (2024) proposed SSD-KD, optimizing distillation with minimal synthetic data. Peng et al. Peng et al. (2022) applied MI to multimodal translation, enabling image-free inference. Despite these advances, challenges remain in ensuring the diversity of generated data for optimal student model performance.

**Gaussian Mixture Model (GMM)** GMMs have advanced in aligning data distributions with target models and capturing diverse characteristics. Reynolds Reynolds (2009) and McLachlan et al. McLachlan & Rathnayake (2014) reviewed GMMs, highlighting their flexibility in modeling data as a mixture of Gaussians. Montesuma et al. Montesuma et al. (2024) leveraged GMMs for domain adaptation via optimal transport, improving distribution alignment. Fuchs et al. Fuchs et al. (2022) introduced MDGMM for clustering mixed datasets, effectively handling heterogeneous data. Li et al. Li et al. (2024) proposed methods to better model rare events within GMMs, enhancing representation of infrequent occurrences. Ben-Yosef and Weinshall Ben-Yosef & Weinshall (2018) developed GM-GAN, combining GMMs with GANs to model complex multimodal distributions. These studies collectively demonstrate the versatility of GMMs in capturing diverse data patterns across various domains.

**Contrastive Learning** Contrastive learning increasingly emphasizes representation diversity to enhance robustness and generalization. Zhou et al. Zhou et al. (2023) proposed Geometric Harmonization to improve category-level uniformity in long-tailed self-supervised learning. CODER Author & Author (2023a) introduced a diversity-sensitive loss for cross-modal contrastive learning, capturing a broader range of features. Research on pretraining data Author & Author (2023b) showed that greater diversity improves self-supervised model performance. These studies highlight the crucial role of diversity in contrastive learning, enabling more flexible and comprehensive modeling.

### A.10 LIMITATIONS

Although the proposed method achieves strong performance, there is still room for further improvement. The current framework integrates both Distributional Recovery and Dual Contrastive Learning within a unified training process, which effectively enhances data quality and model robustness. However, the presence of multiple components and objectives inevitably introduces some additional computational overhead. Future work could explore more lightweight designs or unified objective functions to further improve training efficiency without sacrificing performance. In addition, like most existing Data-Free Knowledge Distillation (DFKD) approaches, including the baselines considered in this study, our experiments primarily focus on image datasets such as CIFAR-10. Extending the proposed method to other data modalities, including text and tabular data, is a promising direction for future research to broaden its applicability and impact.

### A.11 DETAILS OF DATASETS AND MODELS

#### A.11.1 DATASETS

- **CIFAR-10** Krizhevsky et al. (2009): This dataset consists of 60,000 color images divided into 10 classes, with 50,000 training samples and 10,000 test samples. Each image has a resolution of $32 \times 32$ pixels. The classes cover various objects such as airplanes, automobiles, birds, cats, and dogs.

- **CIFAR-100** Krizhevsky et al. (2009): An extension of CIFAR-10, CIFAR-100 includes 100 classes grouped into 20 superclasses. Each class contains 500 training samples and 100 test samples. The image resolution remains at $32 \times 32$ pixels. The larger number of classes and lower number of samples per class make this dataset more challenging than CIFAR-10.

- **Tiny-ImageNet** Le & Yang (2015): A subset of the ImageNet dataset, Tiny-ImageNet contains 200 classes with 500 training images, 50 validation images, and 50 test images per class. All images are resized to $64 \times 64$ pixels. This dataset provides a more complex and diverse set of visual concepts, making it suitable for evaluating the generalization capability of knowledge distillation methods.

#### A.11.2 MODELS

- **ResNet** He et al. (2016): Residual Networks (ResNet) introduce shortcut connections to mitigate the vanishing gradient problem, enabling the training of deep networks. We employ ResNet-34 as the teacher model and ResNet-18 as the student model to evaluate performance on CIFAR-10 and CIFAR-100 datasets.

- **VGG** Simonyan (2014): VGG networks utilize deep architectures with stacked convolutional layers of small receptive fields ($3 \times 3$ kernels). In our experiments, we use VGG-11 as the teacher model and VGG-13 as the student model for the CIFAR-10 dataset evaluation.
- **Wide ResNet (WRN)** Zagoruyko (2016): Wide ResNet enhances the original ResNet by increasing the network width while reducing depth, improving both accuracy and training efficiency. We adopt WRN-40-2 as the teacher model and WRN-16-2 as the student model for evaluations on CIFAR-10, CIFAR-100, and Tiny-ImageNet datasets.

### A.12 MORE BASELINE COMPARISON

To further broaden our empirical comparison, we additionally evaluate D2CIP against the recent NAYER method Tran et al. (2024) under multiple Teacher–Student configurations. As shown in Table 13, D2CIP consistently achieves equal or better performance across CIFAR-10, CIFAR-100, and Tiny-ImageNet. The advantage of D2CIP holds regardless of whether the teacher–student pairs are based on ResNet-34 or ResNet-18, indicating that the improvements are stable across different backbone capacities. These results demonstrate that D2CIP provides more reliable synthetic data quality and stronger distillation effectiveness, further reinforcing its generality and robustness compared with recent alternatives.

Table 13: Performance comparison of D2CIP and NAYER (Tran et al. (2024)) across multiple Teacher–Student ($\mathcal{T} - \mathcal{S}$) pairs. Teacher/Student models include (a) ResNet-34, (b) ResNet-18. The **bold** highlights the best performance within each configuration.

| Method | CIFAR-10 | | CIFAR-100 | | Tiny-ImageNet |
|---|---|---|---|---|---|
| | $\mathcal{T}$(a) $\mathcal{S}$(b) | $\mathcal{T}$(c) $\mathcal{S}$(b) | $\mathcal{T}$(a) $\mathcal{S}$(b) | $\mathcal{T}$(c) $\mathcal{S}$(b) | $\mathcal{T}$(a) $\mathcal{S}$(b) |
| Teacher Model ($\mathcal{T}$) | 95.70 | 95.20 | 78.05 | 77.10 | 63.27 |
| Student Model ($\mathcal{S}$) | 95.20 | 92.25 | 77.10 | 71.32 | 61.37 |
| NAYER | 94.76 | 91.34 | 76.98 | 71.61 | 61.12 |
| D2CIP | **95.16** | **91.70** | **77.36** | **71.66** | **61.49** |

