# OpenReview forum: "Dual Contrastive Inversion with Distributional Priors for Diversity-Aware Data-Free Knowledge Distillation"
_ICLR.cc/2026/Conference — Submitted to ICLR 2026_

### Official Review · Reviewer_GRGG · 2025-10-27

**Soundness:** 3
**Presentation:** 3
**Contribution:** 2
**Rating:** 4
**Confidence:** 3

**Summary:**

The paper proposes a new framework, D2CIP, to enhance data diversity in data-free knowledge distillation (DFKD) by addressing the limitations of existing model inversion (MI) approaches that rely on unimodal priors and lack separability constraints. D2CIP introduces a two-stage process: (1) Distributional Prior Recovery, which learns a class-conditional GMM aligned with the teacher’s predictions and batch-normalization statistics to capture multimodal data structure; and (2) Dual Contrastive Inversion, which applies contrastive learning at both latent and instance levels using memory banks to enlarge negative samples and encourage diversity. Theoretically, the paper formalizes data diversity as expected pairwise separability and proves its monotonic relation with contrastive loss, providing a principled basis for diversity maximization. Experiments on CIFAR-10, CIFAR-100, Tiny-ImageNet, and ImageNet-100 show that D2CIP outperforms previous DFKD methods (e.g., CMI, TA-DFKD) in both synthetic data diversity (lower FID and JS divergence) and distillation accuracy, confirming that its combination of distributional priors and dual contrastive learning yields robust and generalizable data-free distillation

**Strengths:**

1. The use of a class-conditional Gaussian Mixture Model (cGMM) captures multimodal latent structures aligned with teacher statistics, addressing mode collapse; ablation results show large accuracy drops when this component is removed (–1.0% on CIFAR-10, –3.8% on CIFAR-100). This means it’s indeed important to model different classes separately.
2. Quantitative evaluations show consistent superiority in both JS divergence (0.498 vs 0.531) and FID (110.56 vs 112.32) over CMI and TA-DFKD, confirming that the dual contrastive design leads to more diverse and semantically aligned synthetic data.

**Weaknesses:**

1. While the paper presents a well-organized framework, several core components, such as contrastive learning, batch normalization alignment, and adversarial distillation, are closely related to existing works like CMI and DeepInversion. As a result, the boundary between adaptation and innovation could be articulated more clearly to highlight the unique conceptual advances of D2CIP.
2. The paper provides strong quantitative evidence through FID and JS divergence but includes few visual examples of the generated samples. Additional visual comparisons would help readers better appreciate whether D2CIP produces more realistic and diverse images than existing methods.
3. In Figure 3, the generated distributions of D2CIP and CMI appear visually similar, suggesting comparable data manifolds. A more detailed qualitative analysis or higher-resolution visualization could better illustrate the claimed improvements in diversity and distributional richness.

**Questions:**

Please see the weaknesses.

---

> ### Author Response · Authors · 2025-11-26
>
> Thank you for your valuable comments. We will address each weakness as listed below.
>
> >**[W1]** We appreciate the reviewer’s observation. While D2CIP follows the standard gMI framework and therefore incorporates common components such as BN alignment and adversarial distillation, **our contributions lie in addressing limitations** that CMI, DeepInversion, and related works do not tackle.
> >  &nbsp;
> >**First**, **prior methods all rely on fixed or unimodal priors** (random noise, a single Gaussian). D2CIP is the **first to introduce a learnable class-conditional multi-modal prior (cGMM)** jointly optimized with inversion. This provides a much more expressive latent structure and enables recovering class-specific modes rather than optimizing from noise.
> >  &nbsp;
> >**Second**, although CMI uses an output-level contrastive loss, existing methods **lack a principled way to enforce diversity or instance separability**. D2CIP introduces **dual contrastive learning, in both latent and sample space**, with memory banks and provides a theoretical link between contrastive loss and diversity maximization. This theoretical grounding and latent-level design are absent in prior work.
> >  &nbsp;
> >**Third**, ***combining a learnable multi-modal prior with dual contrastive objectives yields a qualitatively different inversion paradigm*** that improves diversity, mode coverage, and distillation performance beyond CMI and DeepInversion. We have **clarified these conceptual distinctions in the revised manuscript**, and the detailed discussion has been added to **Appendix A.2.6 (second paragraph)**.
>
> >**[W2]** We appreciate the reviewer’s suggestion. However, we would like to clarify an important aspect of data-free generation. **Unlike conventional generative modeling, data-free inversion methods do not aim to reconstruct visually realistic real images**, as the generator has no access to real samples or real image priors. Instead, **the objective is to produce feature-valid synthetic data** that effectively activates the teacher model and matches its underlying distribution. For this reason, visual inspection is inherently limited; that is, the synthetic samples are optimized for **distributional fidelity rather than human-perceived realism**, and diversity is often not directly observable by eye.
> >  &nbsp;
> >Consequently, **quantitative metrics such as FID, JS divergence, MMD, and LPIPS** (Table 10, Appendix A.5.2) provide a far **more objective and reproducible evaluation of synthetic quality** in the data-free setting. In addition, we include representative visual comparisons in Figure 11 of Appendix A.7. These examples illustrate the qualitative differences among methods, but the **primary conclusions are supported by the quantitative divergence and diversity metrics**.
>
> >**[W3]** Figure 3 provides a **general distributional visualization** obtained through a 2D dimensionality-reduction method. Such projections are helpful for illustrating overall structural tendencies but, by construction, **do not preserve all characteristics of the original high-dimensional distributions**. Therefore, for an objective comparison of distributional differences, we complement these visualizations with numerous quantitative metrics.
> >  &nbsp;
> >To more **objectively assess distributional richness and diversity**, our paper reports **multiple quantitative metrics, including FID and JS divergence** in the main text, as well as **MMD and LPIPS in Table 10 of Appendix A.5.2**. These metrics are widely used in data-free generation and directly evaluate alignment, mode coverage, and perceptual diversity, and across all of them **D2CIP consistently outperforms CMI**. The **downstream distillation improvements** (Table 2) further demonstrate that the enhanced diversity produced by D2CIP leads to more informative synthetic data rather than superficial similarity in low-dimensional projections.
> >  &nbsp;
> >In summary, **consistent with standard practice** in the DFKD literature, **the combination of quantitative divergence metrics and downstream distillation performance provides a sufficient and objective evaluation of diversity and distributional richness**. These results already demonstrate that D2CIP achieves clearly superior diversity and distributional fidelity compared with CMI and other SOTA baselines.
>
> Please consider **raising your score** if our response addresses your concerns.

---

### Official Review · Reviewer_YNcZ · 2025-10-31

**Soundness:** 3
**Presentation:** 3
**Contribution:** 2
**Rating:** 4
**Confidence:** 5

**Summary:**

The paper proposes a novel data-free knowledge distillation method that controls the diversity and quality of synthetic images using a Gaussian Mixture Model for input generation instead of random noise. It also introduces a dual contrastive inversion strategy that applies contrastive learning at both latent and instance levels, supported by memory banks to expand the negative sample space.

**Strengths:**

1. The paper is well written and easy to follow.
2. The idea of using a prior distribution to generate inputs for the generator is sound and clearly explains why it can improve performance.

**Weaknesses:**

1. The main novelty of this paper lies in controlling the diversity and quality of the generator’s outputs by using a prior distribution to create the input, rather than relying on random noise. However, this idea is quite similar to the approach in [1], where fixed inputs (based on label text embeddings) are used, and randomness is introduced through additional layers. It would be beneficial if the authors could discuss the advantages and disadvantages of these two techniques.
2. The experiments lack comparisons with some state-of-the-art methods, such as [1] and [2].
3. The paper reports results only on a small dataset and focuses solely on the image classification task.

[1] Nayer: Noisy layer data generation for efficient and effective data-free knowledge distillation. CVPR 2025.
[2] Coupling the Generator with Teacher for Effective Data-Free Knowledge Distillation. ICCV 2025.

**Questions:**

Please see the weakness section. I will consider raising my score if Question 1 is clearly explained.

---

> ### Author Response · Authors · 2025-11-26
>
> Thank you for your valuable comments. We will address each weakness as listed below.
> >
> >**[W1]** Although both papers aim to improve controllability and diversity in data-free generation, our method is **conceptually and technically different from NAYER** in several fundamental aspects.
> >  &nbsp;
> >**First**, the **nature of the generator’s input** differs substantially. **NAYER** replaces random noise with a **fixed label–text embedding (LTE)**, giving each class only one deterministic vector that **never changes**. In contrast, **D2CIP** includes a **learnable class-conditional distributional prior in the form of a cGMM**, whose parameters are optimized together with the generator under teacher predictions and BN statistics. This **prior is multi-modal, class-specific, and adaptive**, allowing it to represent an evolving manifold rather than a single point and to **capture substantially richer intra-class structure**.
> >  &nbsp;
> >**Second**, the **mechanisms for achieving diversity** are fundamentally different. **NAYER** injects randomness by reinitializing a shallow noisy layer that perturbs the fixed LTE, which yields **limited variation and no explicit control over instance separability**. **Our method** provides two principled and complementary sources of diversity:
> (1) ***distributional diversity from the multi-modal cGMM prior***, enabling exploration of multiple latent modes; and
> (2) ***contrastive diversity via dual contrastive learning on both latent vectors and generated samples***, with memory banks expanding the pool of negatives.
> We further show that minimizing the contrastive loss monotonically increases expected pairwise separability, giving us a **theoretically grounded objective absent in NAYER**.
> >  &nbsp;
> >**Third**, the **optimization objectives** differ in intent. **NAYER** follows standard CE, BN, adversarial losses, but **lacks mechanisms to enforce instance-level discrimination or prevent collapse** into a narrow set of modes. In contrast, **our framework ** explicitly incorporates **dual contrastive losses to promote both inter-instance separability and inter-mode coverage**, while maintaining class correctness and BN alignment. This substantially improves diversity and contributes directly to higher distillation accuracy.
> >  &nbsp;
> >We acknowledge that learning a distributional prior introduces modest computational overhead compared to using fixed textual embeddings. However, this is a deliberate trade-off that enables the model to learn richer intra-class structure and achieve significantly higher diversity and distillation performance.
> >  &nbsp;
> >In summary, while both approaches seek controllable generation, **NAYER is constrained by fixed embeddings and architecture-level noise**, limiting it to a single deterministic input per class. **Our method instead employs a learnable multi-modal prior and dual contrastive objectives that explicitly optimize diversity and separability**, yielding a fundamentally more expressive and theoretically grounded generation paradigm.
>
> >**[W2]** As for paper [1] (CVPR 2024), we have **added additional experiments in Appendix A.12** to address this concern. Specifically, we evaluate D2CIP against the recent NAYER method under multiple T–S configurations on CIFAR-10, CIFAR-100, and Tiny-ImageNet. As shown in **Table 13 of Appendix A.12**, ***D2CIP consistently matches or surpasses NAYER across all datasets and T–S pairs***. These results indicate that D2CIP produces more reliable synthetic data and stronger distillation performance, reinforcing its robustness and generality compared with this state-of-the-art alternative.
> >  &nbsp;
> >Regarding paper [2] (ICCV 2025), we note that it was **released in October 2025**, which is **after the full-paper submission deadline for ICLR 2026 (24 September 2025)**. This is the reason it was not included among our baselines. Following the reviewer’s suggestion, we will add this method as an additional baseline in the camera-ready version.
>
> >**[W3]** Regarding the **choice of datasets**, we emphasize that **our experimental settings**, including the use of CIFAR-10, CIFAR-100, and Tiny-ImageNet, **are fully consistent with standard practice** in the data-free knowledge distillation literature. All representative SOTA baselines, such as ***DFQ (ICCV 2019), CMI (IJCAI 2021), PRE-DFKD (AAAI 2022), LS-GDFD (ICCV 2023), and TA-DFKD (AAAI 2024)***, evaluate under the same dataset protocol, as do the most recent works [1, 2] cited by the reviewer. **This ensures strict comparability across methods**.
> >  &nbsp;
> >Furthermore, to **enhance the comprehensiveness of our evaluation**, we additionally **extend experiments** to a larger and more challenging setting, **ImageNet, reported in Appendix A.3**. These results demonstrate that D2CIP remains effective beyond small-scale datasets.
>
> Please consider **raising your score** if our response addresses your concerns.

---

> > ### Comment · Reviewer_YNcZ · 2025-11-28
> >
> > Thank you for your response.
> >
> > > [W2] As for paper [1] (CVPR 2024), we have added additional experiments in Appendix A.12 to address this concern. Specifically, we evaluate D2CIP against the recent NAYER method under multiple T–S configurations on CIFAR-10, CIFAR-100, and Tiny-ImageNet. As shown in Table 13 of Appendix A.12, D2CIP consistently matches or surpasses NAYER across all datasets and T–S pairs. These results indicate that D2CIP produces more reliable synthetic data and stronger distillation performance, reinforcing its robustness and generality compared with this state-of-the-art alternative.
> >
> > #### The results reported for NAYER in Appendix A.12 appear inconsistent with those in [1]. For example, for R34–R18 on CIFAR-10, you report D2CLIP at 95.16 and NAYER at 94.76, whereas [1] reports 95.21 for NAYER. Similar discrepancies occur in other comparisons, where your NAYER results are noticeably lower than those in [1]. This raises a serious concern about the claimed benefits of the method and whether D2CIP can truly be regarded as state-of-the-art DFKD.
> >
> > > [W3] Regarding the choice of datasets, we emphasize that our experimental settings, including the use of CIFAR-10, CIFAR-100, and Tiny-ImageNet, are fully consistent with standard practice in the data-free knowledge distillation literature. All representative SOTA baselines, such as DFQ (ICCV 2019), CMI (IJCAI 2021), PRE-DFKD (AAAI 2022), LS-GDFD (ICCV 2023), and TA-DFKD (AAAI 2024), evaluate under the same dataset protocol, as do the most recent works [1, 2] cited by the reviewer. This ensures strict comparability across methods.
> >
> > #### Nevertheless, the paper would be more convincing if the proposed approach were also demonstrated on more challenging or diverse tasks, such as full ImageNet, data-free quantization, or segmentation. I understand that, due to time constraints, it may be difficult to conduct additional large-scale experiments, but even partial results or a more detailed discussion of the method’s applicability in these settings would further strengthen the paper.

---

### Official Review · Reviewer_fST2 · 2025-10-31

**Soundness:** 3
**Presentation:** 3
**Contribution:** 2
**Rating:** 4
**Confidence:** 3

**Summary:**

This paper points out that the existing model inversion-based data-free knowledge distillation suffers limited synthetic data diversity problem, which from unimodal prior and implicit mechanisms for instance separability. To solve this issue, this paper proposes a two-stage framework D2CIP to enhance the diversity of synthetic data, which includes Distributional Prior Recovery and Dual Contrastive Inversion stages. For the first stage, Distributional Prior Recovery, it learns a class-conditional prior and adopt GMM to jointly optimized with the generator to align with the predictions of the teacher and batch-normalization statics, which can capture the multi-modal nature of real data. In second stage, it applies dual contrastive learning at both latent and instance levels, and adopts memory banks to expand negative samples, enhancing instance discrimination and maximizing data diversity. It further defines data diversity as expected pairwise separability. For downstream distillation, it proposes a decision-adversarial strategy for boundary samples. The experimental results demonstrate the effectiveness of the proposed method.

**Strengths:**

1.	This paper is well written with clear and illustrative figures and tables, and well-motivated.
2.	This paper defines data diversity as expected pairwise separability and theoretically proves the monotonic relationship with contrastive loss.
3.	The experimental results demonstrate the SOTA performance of the proposed method on CIFAR-10, CIFAR-100, Tiny-ImageNet datasets.

**Weaknesses:**

1.	[major] The paper introduces key components, e.g., the class-conditional GMM prior, dual contrastive objectives, and memory banks, which improve the performance of the synthetic data for downstream distillation. However, they also lead to a notable increase in training time and GPU memory usage as increased learnable parameters. As shown in Table 11, D2CIP requires 1.5 times higher memory usage and longer training time per epoch compared with baseline. Although, the paper reports only the comparison of runtime and memory requirement per epoch, without further analysis of the total number of epochs required for training convergence and the overall training cost, it is clear that the proposed framework results in an increase in computational resource requirements.
2.	[minor] Due to the major weakness in computational efficiency, the proposed method may have scaling-up problems, for larger datasets, e.g., ImageNet-1K, or more complex teacher-student architectures.
3.	[major] The proposed framework introduces too many learnable or tunable parameters but does not clarify how these values are determined or how they should be tuned for different datasets or architectures.
4.	[minor] All experiments are conducted on CNN-based architectures, ResNet, VGG, WRN, and the method has not been evaluated on more diverse or modern backbones such as Vision Transformers.

**Questions:**

1.	As mentioned in weakness [major] 1, this paper (Table 11) only reports runtime and memory usage per epoch. Could the authors provide the total training cost, including number of epochs to convergence and total GPU hours, to better support the overall efficiency of D2CIP?
2.	The performance appears sensitive to parameter choices, could the authors provide clearer guidance or for parameter setting without extensive tuning?
3.	Could the proposed method apply on larger-scale dataset, e.g., ImageNet-1K, and more complex architectures, e.g., ViT?

---

> ### Author Response · Authors · 2025-11-26
>
> Thank you for your valuable comments. We will address each weakness as listed below.
> >
> > **[W1]** **First**, we acknowledge that the newly introduced components in D2CIP (multi-modal prior, dual contrastive losses, and memory banks) introduce a moderate increase in computational overhead, ***primarily due to the memory banks***, as discussed in Appendix A.6. However, the overall training cost of D2CIP is substantially mitigated by its faster and more stable convergence, as demonstrated by the **newly provided training loss curves in Figure 11 of Appendix A.6** in the revised manuscript. As shown in Figure 11, D2CIP maintains a **consistently lower and more stable training loss trajectory throughout training**, whereas TA-DFKD and PRE-DFKD exhibit slower descent, larger oscillations, or higher plateaued losses. Since the learnable multi-modal prior provides a more expressive latent structure and the dual contrastive objectives prevent mode collapse, the generator **achieves stable and diverse samples** much earlier than the baselines.
> >  &nbsp;
> >**Second**, in data-free knowledge distillation, ***the primary objective is to achieve higher diversity, better distributional fidelity, and stronger downstream distillation performance***. Our experimental results clearly show that the additional computation introduced by D2CIP is well justified, as it yields consistent performance gains across all benchmarks.
> >  &nbsp;
> >Taken together, these results demonstrate that the overall computational cost of D2CIP remains **practical**, and the moderate overhead represents a **reasonable trade-off for the better improvements in diversity and distillation effectiveness**.
>
> >
> > **[W2]** We **respectfully disagree with the concern that D2CIP cannot scale**. Our results already demonstrate that the method is practical on significantly larger datasets and is not fundamentally limited by its added components.
> >  &nbsp;
> >**First**, as shown in **Table 4 of Appendix A.3**, we successfully evaluate D2CIP on the **larger ImageNet-100 benchmark** (a standard 100-class subset of ImageNet-1K for efficient large-scale evaluation). While most existing DFKD baselines, such as ***CMI (IJCAI 2021), PRE-DFKD (AAAI 2022), and TA-DFKD (AAAI 2024)***, are unable to report results on this dataset due to computational constraints, D2CIP achieves 51.98% **Top-1 accuracy, outperforming** CMI (41.04%), PRE-DFKD (43.36%), and TA-DFKD (33.66%). This demonstrates that **D2CIP scales effectively to more demanding and larger settings**.
> >  &nbsp;
> >**Second**, the reviewer’s concern regarding the **use of more complex teacher–student architectures is not inherent to D2CIP**. Our method generates synthetic data using only the predictive capability of the teacher model; it **does not rely on architectural details** such as depth, parameter count, or backbone type. **As long as the teacher provides sufficiently reliable logits**, which is typically the case for modern high-capacity models, the generator training remains stable, and the effectiveness of D2CIP **is unaffected by the specific T/S architecture**.
> >  &nbsp;
> > **Furthermore**, as per your suggestion, we will include additional experiments using ViT teachers.
>
> > **[W3]** Although D2CIP introduces several additional components, **the number of actual tunable hyperparameters remains comparable to, or even fewer than**, those used in prior model inversion and data-free KD methods.
> >  &nbsp;
> > **First**, **most of our hyperparameters are loss-balancing coefficients**, ***consistent with standard practices*** in ***CMI (IJCAI 2021), PRE-DFKD (AAAI 2022), and LS-GDFD (ICCV 2023)***. These baselines already rely on multiple tunable losses (e.g., CE, BN, adversarial mismatch). D2CIP introduces **only two weights for the diversity objectives and two for learning the latent prior**, each directly corresponding to the specific limitations we address (simplistic priors and insufficient instance diversity). Their values are chosen by ensuring that each loss term remains on a comparable scale, making them straightforward to set and transfer across datasets.
> >  &nbsp;
> >**Second**, the improvements in distributional richness, diversity metrics, and downstream distillation accuracy confirm that **each additional term serves a clear functional purpose**. We also provide **a detailed hyperparameter sensitivity study in Appendix A.4**, ***offering concrete guidance for practitioners on adapting D2CIP to new datasets and architectures***.

---

> ### Author Response · Authors · 2025-11-26
>
> > **[W4]** We clarify that our **experimental setup follows the standard and widely adopted evaluation protocol** in the DFKD literature.
> >  &nbsp;
> >**First**, our choice of architectures (ResNet, VGG, WRN) **is fully aligned with representative SOTA baselines**, including ***DFQ (ICCV’19), CMI (IJCAI’21), PRE-DFKD (AAAI’22), LS-GDFD (ICCV’23), and TA-DFKD (AAAI’24)***. These works consistently evaluate on CNN-based backbones using identical teacher–student pairs (see Table 2), and we adopt the **same protocol to ensure strict comparability and fair benchmarking**.
> >  &nbsp;
> >**Second**, incorporating Vision Transformers **does not affect the validity or generality of our pipeline**. D2CIP operates entirely in the latent-prior and model-inversion space and is agnostic to the architectural family of the teacher (T) and student (S). **As long as the teacher provides reliable logits**, the generator optimization and the proposed contrastive objectives remain unchanged. Thus, ***extending to ViTs is orthogonal to the conceptual design of D2CIP***.
> >  &nbsp;
> >**Finally**, as shown in Appendix A.3, D2CIP **already scales to larger datasets such as ImageNet-100**, demonstrating its applicability beyond small-scale settings. Following the reviewer’s suggestion, we will include additional experiments with ViT teachers in the camera-ready version to further validate the generality of D2CIP.
>
> > **[Q1]** See the reply for **[W1]**.
>
> > **[Q2]** See the reply for **[W3]**.
>
> > **[Q3]** See the reply for **[W2]**.
>
> Please consider **raising your score** if our response addresses your concerns.

---

### Meta-Review · Area_Chair_7TM6 · 2025-12-15

**Summary:**

Reviewer fST2 raised main concerns on the increment of training time, GPU memory and learnable parameters and missing experiments on ViT architecture. Reviewer YNcZ pointed out its novelty issue about NAYER and missing comparative experiments. Reviewer GRGG expressed concerns on novelty (e.g., CMI and DeepInversion) and visualization comparison. The AC agrees with Reviewer GRGG for novelty issues of the submission. The contributions relative to CMI and DeepInversion are somewhat limited, including contrastive objective, adversarial training and batch normalization. Hence, it's recommended as rejection.
Moreover, the code link should not be invalid during review.

**Reviewer Concerns:**

The concerns about the novelty issue from Reviewer YNcZ have been well addressed.

The concerns about overhead consumption and novelty issue relative to CMI and DeepInversion still exist.

**Reviewer Scores:**

Reviewer fST2: 4

Reviewer YNcZ: 4

Reviewer GRGG: 4

---

### Decision · Program_Chairs · 2026-01-26

Reject